# Multivariate Latent Recalibration for Conditional Normalizing Flows

**Victor Dheur**
Department of Computer Science
University of Mons
Mons, Belgium
victor.dheur@umons.ac.be

**Souhaib Ben Taieb**[*]
Department of Statistics and Data Science
Mohamed bin Zayed University of Artificial Intelligence
Abu Dhabi, United Arab Emirates
souhaib.bentaieb@mbzuai.ac.ae

## Abstract

A reliable estimate of the full conditional distribution of a multivariate response given a set of covariates is essential in many decision-making applications. However, misspecified or miscalibrated models can lead to poor approximations of the joint distribution, resulting in unreliable predictions and suboptimal decisions. Standard recalibration methods are largely restricted to univariate settings, and while conformal prediction techniques yield multivariate regions with coverage guarantees, they do not provide an explicit form of the underlying probability distribution. We address this gap by first introducing a novel notion of latent calibration, which assesses probabilistic calibration in the latent space of conditional invertible generative models such as normalizing flows and flow matching. Second, we propose latent recalibration (LR), a post-hoc model recalibration method that learns a transformation of the latent space with finite-sample bounds on latent calibration. Unlike existing recalibration methods, LR produces a recalibrated distribution with an explicit multivariate density function while remaining computationally efficient. Extensive experiments on both tabular and image datasets show that LR consistently improves latent calibration error and the negative log-likelihood of the recalibrated models.

## 1 Introduction

Generating reliable uncertainty estimates is essential for trustworthy decision-making across a wide range of applications (Gawlikowski et al., 2023). Multi-output regression problems, in particular, arise frequently in domains such as weather forecasting (Setiawan et al., 2024), energy consumption prediction (Makaremi, 2025), and healthcare resource utilization (Cui et al., 2018). While flexible models like neural networks can achieve high predictive accuracy, their uncertainty estimates are often poorly *calibrated*, meaning predicted probabilities or confidence regions do not align with empirical frequencies (Guo et al., 2017; Dheur and Ben Taieb, 2023). Furthermore, most recalibration methods are designed for the single-output setting (Gneiting et al., 2007; Song et al., 2019; Sahoo et al., 2021; Kuleshov and Deshpande, 2021; Dewolf et al., 2022; Fakoor et al., 2023; Marx et al., 2023; Chung et al., 2023; Gneiting and Resin, 2023).

Noting the general lack of methods for *assessing* and *recalibrating* multi-output models, Chung et al. (2024) leveraged highest-density regions (HDRs) (Hyndman, 1996) to introduce the notion of HDR calibration and propose the sampling-based HDR recalibration (HDR-R) method. Recently, multi-output conformal prediction (CP) methods have also been developed to construct joint prediction sets (Wang et al., 2023; Feldman et al., 2023; Fang et al., 2025; Dheur et al., 2025). However, both HDR-R and CP approaches fail to provide an explicit form for the underlying recalibrated probability

---

[*]Also affiliated with the Department of Computer Science, University of Mons.

39th Conference on Neural Information Processing Systems (NeurIPS 2025).

distribution, with HDR-R further involving computationally intensive sampling and binning at test time.

To overcome these limitations, we introduce the *latent recalibration (LR)* method, based on a new notion of *latent calibration*, which recalibrates invertible generative models (e.g., normalizing flows (NFs) or flow matching (FM)) by operating within their latent space. The core idea is to learn a transformation of the latent space such that the resulting model achieves latent calibration. Compared to set-based CP methods and sampling-based recalibration approaches such as HDR-R, LR (i) yields a fully recalibrated generative model with an explicit multivariate probability density function (PDF), (ii) provides finite-sample latent calibration guarantees, and (iii) enables efficient density evaluation and sampling, which are essential for many applications and support improved decision-making (Klein, 2024).

Our main contributions are:

- We introduce latent calibration, a new notion of calibration evaluated within the latent space of an invertible generative model, based on the distribution of latent norms.

- We propose LR, a recalibration method that yields a multivariate predictive distribution with an explicit PDF, offers finite-sample latent calibration guarantees, and remains computationally efficient.

- We empirically demonstrate, across 29 multi-output tabular datasets and one high-dimensional image dataset, that LR consistently improves latent calibration and reduces negative log-likelihood (NLL). A public codebase is provided to ensure reproducibility.[1]

## 2 Background

We consider a multi-output distributional regression setting, where the goal is to predict the distribution of a $d$-dimensional response variable $Y \in \mathcal{Y} \subseteq \mathbb{R}^d$ from a $p$-dimensional input vector $X \in \mathcal{X} \subseteq \mathbb{R}^p$. We assume access to a dataset $\mathcal{D} = \{(X^{(j)}, Y^{(j)})\}_{j=1}^N$, where the samples $(X^{(j)}, Y^{(j)})$ are drawn i.i.d. from $P_{XY}$, the true joint distribution over $\mathcal{X} \times \mathcal{Y}$. The dataset is partitioned into three disjoint subsets: a training set $\mathcal{D}_{\text{train}}$, a test set $\mathcal{D}_{\text{test}}$, and a calibration set $\mathcal{D}_{\text{cal}} = \{(X^{(i)}, Y^{(i)})\}_{i=1}^n$. The true conditional distribution of $Y$ given $X = x$ is denoted by $P_{Y|X=x}$. Similarly the true cumulative distribution function (CDF) is denoted $F_{Y|X=x}$, and the corresponding PDF, assumed to exist for all $x \in \mathcal{X}$, is denoted $f_{Y|X=x}$. More generally, we denote the distribution, CDF and PDF of any random variable $A$ by $P_A$, $F_A$, and $f_A$, respectively. Additionally, estimates are denoted $\hat{P}_A$, $\hat{F}_A$, and $\hat{f}_A$.

### 2.1 Normalizing Flows for predictive density estimation

Using $\mathcal{D}_{\text{train}}$, our goal is to estimate the conditional PDF $f_{Y|X=x}$ for inputs $x \in \mathcal{X}$. NFs offer a flexible framework for modeling complex distributions over continuous random variables. Specifically, an NF defines a learnable bijective (conditional) transformation $\hat{T} : \mathcal{Z} \times \mathcal{X} \to \mathcal{Y}$ between a latent space $\mathcal{Z} \subseteq \mathbb{R}^d$ and the output space $\mathcal{Y} \subseteq \mathbb{R}^d$. For any $y \in \mathcal{Y}$ and $x \in \mathcal{X}$, the transformation satisfies $\hat{T}(\hat{T}^{-1}(y; x); x) = y$. Given an input $x \in \mathcal{X}$, the NF maps a latent random variable $Z \in \mathcal{Z}$ (typically drawn from a known base distribution, such as $\mathcal{N}(0, I_d)$) to a new random variable $\hat{T}(Z; x)$. The resulting conditional PDF is computed using the change-of-variables formula:

$$\hat{f}_{Y|X=x}(y) = f_Z(\hat{T}^{-1}(y; x)) \left| \det \left( \nabla_y \hat{T}^{-1}(y; x) \right) \right|, \tag{1}$$

where $f_Z$ denotes the density of the latent variable $Z$. NFs are typically trained by minimizing the NLL over the training dataset using mini-batch stochastic gradient descent. For details on NF architectures, we refer the reader to Papamakarios et al., 2021.

For brevity, the main text focuses on normalizing flows, but our method is also compatible with flow matching, as shown in Section L.

---

[1] https://github.com/Vekteur/latent-recalibration

## 2.2 Statistical calibration

While training with strictly proper scoring rules such as the NLL encourages accurate predictions, it does not guarantee that the resulting predictions are reliable or calibrated, meaning they are statistically aligned with the true distribution of the observations (Gneiting et al., 2007). This issue is particularly relevant under limited data or model misspecification, and it has gained renewed attention with the observation that modern neural network classifiers are often miscalibrated and overconfident (Guo et al., 2017).

**Probabilistic calibration.** For real-valued outcomes ($d = 1$), probabilistic calibration (Gneiting et al., 2007) builds on the probability integral transform (PIT). Denote $\hat{F}_{Y|X}$ a predictive CDF for $Y$ whose value depends on the random variable $X$. Assuming $\hat{F}_{Y|X=x}$ is continuous for any $x \in \mathcal{X}$, probabilistic calibration requires that

$$F_{Y|X}(Y) \sim \mathcal{U}(0, 1). \tag{2}$$

**Multivariate calibration.** Compared to the univariate case, calibration for vector-valued outcomes has been relatively underexplored. Moreover, assessing calibration in the multivariate setting ($d \geq 2$) is inherently more challenging, as the PIT is no longer uniformly distributed (Genest and Rivest, 2001). While probabilistic calibration can be assessed separately for each dimension, this approach may miss important dependencies between outputs (Chung et al., 2024).

The primary approach to assessing multivariate calibration involves reducing multivariate predictions and observations to univariate summary statistics, and then evaluating the uniformity of the PITs of these transformed values (Allen et al., 2024). Let $(X, Y) \sim P_{X,Y}$ and $(X, \hat{Y}) \sim P_X \hat{P}_{Y|X}$, and define a transformation function (also known as a pre-rank function) $g : \mathcal{X} \times \mathcal{Y} \to \mathbb{R}$. If $G = g(X, Y)$ and $\hat{G} = g(X, \hat{Y})$, then by the probability integral transform, the random variable $\hat{U} = F_{\hat{G}}(G)$ is uniformly distributed whenever $\hat{G} \overset{d}{\approx} G$. In this case, we say that $\hat{P}_{Y|X}$ is probabilistically calibrated with respect to the transformation $g$. Chung et al. (2024) introduced HDR calibration as a special case, where $g(x, y) = -\hat{f}_{Y|X=x}(y)$. In that case, the corresponding PIT $\hat{U} = F_{\hat{G}}(G) = \text{HPD}_{\hat{f}_{Y|X}}(Y)$ is the highest predictive density (HPD; Box and Tiao, 1992). This form of calibration ensures that HDRs derived from the predictive distribution achieve correct empirical coverage at all nominal probability levels.

## 2.3 Recalibration methods

Recalibration methods adjust a potentially miscalibrated base predictor (e.g., $\hat{F}_{Y|X}$) to produce an updated predictor (e.g., $\hat{F}'_{Y|X}$) that satisfies a desired calibration property.

**Quantile recalibration.** For the univariate case ($d = 1$), quantile recalibration (QR) (Kuleshov et al., 2018) is a recalibration method that enforces probabilistic calibration. Let $\hat{U} = \hat{F}_{Y|X}(Y)$, and define $F_{\hat{U}}$, the CDF of $\hat{U}$, as the *calibration map*. The recalibrated CDF is given by $\hat{F}'_{Y|X} = F_{\hat{U}} \circ \hat{F}_{Y|X}$. By construction, $\hat{F}'_{Y|X}(Y)$ is uniformly distributed over $[0, 1]$. Specifically, for any $\alpha \in (0, 1)$:

$$\mathbb{P}(\hat{F}'_{Y|X}(Y) \leq \alpha) = \mathbb{P}(\hat{F}_{Y|X}(Y) \leq F_{\hat{U}}^{-1}(\alpha)) = F_{\hat{U}}(F_{\hat{U}}^{-1}(\alpha)) = \alpha. \tag{3}$$

**HDR recalibration.** For $d \geq 1$, Chung et al. (2024) proposed a sampling-based HDR recalibration method (HDR-R) that targets HDR calibration. Let $\hat{U} = F_{\hat{G}}(G)$, and define $F_{\hat{U}}$ as the calibration map. Given a new input $x_{\text{test}}$, $K$ candidates samples $\{\hat{Y}^{(k)}\}_{k=1}^K$ are drawn from $\hat{P}_{Y|X=x_{\text{test}}}$, with pre-rank values $\hat{G}^{(k)} = -\hat{f}_{Y|X=x_{\text{test}}}(\hat{Y}^{(k)})$, $k = 1, \ldots, K$. Based on binning, HDR-R resamples from the set of candidate samples, producing a new set of recalibrated samples $\{\hat{Y}'^{(k)}\}_{k=1}^K$ with pre-rank values $\hat{G}'^{(k)} = -\hat{f}_{Y|X=x_{\text{test}}}(\hat{Y}'^{(k)})$. This resampling process induces a new random variable $\hat{Y}'$ and its pre-rank $\hat{G}' = -\hat{f}_{Y|X=x_{\text{test}}}(\hat{Y}')$. The number of samples in each bin is determined such that

$$F_{\hat{G}'}(-\hat{f}_{Y|X}(Y)) \sim \mathcal{U}(0, 1). \tag{4}$$

Table 1: Comparison of calibration notions, the associated calibration statistic $\hat{U}$ (uniform under calibration), recalibration methods, and related conformal conformity scores.

| Calibration notion | Calibration statistic | Recalibration method | Conformal method |
|---|---|---|---|
| Probabilistic ($d = 1$) | $\hat{F}_{Y\|X}(Y)$ | Quantile recalibration (QR) | DCP |
| HDR ($d \geq 1$) | $\text{HPD}_{\hat{f}_{Y\|X}}(Y)$ | HDR recalibration (HDR-R) | HPD-split |
| **Latent** ($d \geq 1$) | $F_{\rho_{\mathcal{Z}}(Z)}(\ell_{\hat{T}}(Y;X))$ | **Latent recalibration** (LR) | CONTRA/L-CP |

For completeness, we provide an exact algorithm in Section E.2. While effective in calibrating HDRs, HDR-R has several limitations: (i) it does not produce an explicit recalibrated PDF $\hat{f}'_{Y\|X}$; (ii) it generates duplicate samples; (iii) it is subject to discretization errors when estimating $F_{G\|x}$; and (iv) it is computationally expensive, as it requires generating $K$ initial samples for every recalibrated output.

**Estimating the calibration map.**  In practice, the ideal calibration map $F_{\hat{U}}$ is unknown and is estimated based on the calibration statistics $\{\hat{U}_i\}_{i=1}^n$ with $\hat{U}_i \sim F_{\hat{U}}$. A standard estimator is the empirical CDF $\hat{F}_{\hat{U}}(\alpha) = \frac{1}{n}\sum_{i=1}^n \mathbb{1}(\hat{U}_i \leq \alpha)$ but differentiable estimators can also be employed (Marx et al., 2022; Dheur and Ben Taieb, 2024).

## 2.4 Conformal prediction

Conformal prediction (CP) constructs distribution-free prediction sets $\hat{R}_\alpha(X)$ with finite-sample marginal coverage guarantees, i.e., $\mathbb{P}(Y \in \hat{R}_\alpha(X)) \geq 1 - \alpha$ for any desired significance level $\alpha \in (0, 1)$ (Vovk et al., 2005; Angelopoulos and Bates, 2021). A common variant, Split CP (SCP) (Papadopoulos et al., 2002), partitions the data into a training set $\mathcal{D}_{\text{train}}$ and a calibration set $\mathcal{D}_{\text{cal}}$. A base predictor is trained on $\mathcal{D}_{\text{train}}$, and then a conformity score $s : \mathcal{X} \times \mathcal{Y} \to \mathbb{R}$, where lower values indicate better agreement between the model's predictions and the observations. SCP constructs the prediction set by computing the empirical $(1 - \alpha)$-quantile of the conformity scores evaluated on the calibration set, i.e., $\{s(X^{(1)}, Y^{(1)}), \ldots, s(X^{(n)}, Y^{(n)}), +\infty\}$, denoted by $\hat{F}_S^{-1}(1 - \alpha)$. The resulting prediction region is $\hat{R}_\alpha(x) = \{y \in \mathcal{Y} : s(x,y) \leq \hat{F}_S^{-1}(1 - \alpha)\}$, which satisfies the marginal coverage guarantee.

Notably, specific choices of conformity scores correspond to recalibration statistics: Distributional Conformal Prediction (DCP) (Chernozhukov et al., 2021) uses $s_{\text{DCP}}(x,y) = \hat{F}_{Y\|X=x}(y)$, while HPD-split (Izbicki et al., 2022) uses $s_{\text{HPD-split}}(x,y) = \text{HPD}_{\hat{f}_{Y\|X=x}}(y)$; these match the transformations used in QR and HDR-R, respectively. This highlights a unified framework connecting conformal prediction and recalibration, summarised in Table 1.

## 3 A New Latent Recalibration Method for Normalizing Flows

We propose a new recalibration method, called *latent recalibration* (LR), for conditional NFs. LR operates in the latent space and is specifically designed to achieve our newly introduced notion of multivariate *latent calibration*.

### 3.1 A New Notion of Multivariate Latent Calibration

Recall that, given a latent variable $Z \in \mathcal{Z}$ with a known distribution and an input $x \in \mathcal{X}$, conditional NFs estimate the conditional distribution of $Y$, $F_{Y\|X=x}$, by learning a conditional bijective transformation $\hat{T} : \mathcal{Z} \to \mathcal{Y}$ such that the PDF $\hat{f}_{Y\|X}$ of the transformed variable $\hat{T}(Z; x)$ approximates $f_{Y\|X=x}$. However, model misspecification or significant estimation errors in the learned transformation $\hat{T}$ can lead to poor calibration of the induced distribution of $\hat{T}(Z; x)$.

We propose to leverage the simple structure of the latent space $\mathcal{Z}$ and assess calibration directly in this space, a notion we refer to as *latent calibration*. By definition, if the NF is well-specified for $F_{Y\|X=x}$, then the inverse transformation $\hat{T}^{-1}$ satisfies $\hat{T}^{-1}(Y; X) \overset{d}{\approx} Z$, where $\overset{d}{\approx}$ denotes

approximate equality in distribution. Building on this observation, we define a norm $\rho_{\mathcal{Z}} : \mathcal{Z} \to \mathbb{R}_+$ over $\mathcal{Z}$ (e.g., $\rho_{\mathcal{Z}}(z) := \|z\|$). The goal is to test whether $\rho_{\mathcal{Z}}(\hat{T}^{-1}(Y;X)) \stackrel{d}{\approx} \rho_{\mathcal{Z}}(Z)$.

Since the distribution of $Z$ is known and standard (e.g., standard Gaussian), the distribution of $\rho_{\mathcal{Z}}(Z)$ is often known in closed-form. For instance, if $Z \sim \mathcal{N}(0, I_d)$ and $\rho_{\mathcal{Z}}(z) = \|z\|$, then $\rho_{\mathcal{Z}}(Z)$ follows a Chi distribution with $d$ degrees of freedom ($\chi_d$), whose PDF, CDF, and quantile function can be computed efficiently. As another example, if $Z \sim \mathcal{U}(B_d)$ is uniformly distributed over the unit hyperball and $\rho_{\mathcal{Z}}(z) = \|z\|$, then $\rho_{\mathcal{Z}}(Z)$ follows a Beta$(d, 1)$ distribution.

**Definition 1.** Consider a NF defined by a latent variable $Z$ and a bijective transformation $\hat{T}$. For a pair $(X, Y)$, define the *latent norm* w.r.t. $\hat{T}$ as

$$\hat{L} = \ell_{\hat{T}}(Y; X) = \rho_{\mathcal{Z}}(\hat{T}^{-1}(Y; X)). \tag{5}$$

The NF is said to be *latent calibrated* w.r.t. $Z$ and the norm $\rho_{\mathcal{Z}}$ if the PIT of the latent norm follows a standard uniform distribution, i.e.,

$$\hat{U} = F_{\rho_{\mathcal{Z}}(Z)}(\hat{L}) \sim \mathcal{U}(0, 1). \tag{6}$$

To assess whether a model is latent calibrated, we define the *latent expected calibration error* (L-ECE) as the $L^1$ distance between the CDF of the PIT variable $\hat{U}$ and the CDF of the uniform distribution:

$$\text{L-ECE}(\hat{T}) = \int_0^1 \left| F_{\hat{U}}(\alpha) - \alpha \right| \, d\alpha, \tag{7}$$

The L-ECE is minimized at 0 when $\hat{T}$ is perfectly latent calibrated, and has a maximum value of 0.5.

## 3.2 Multivariate Latent Recalibration

We propose a multivariate latent recalibration method, called LR, which performs a post-hoc adjustment of the latent space of a NF to ensure that the resulting model is latent calibrated. Key advantages of LR are that it yields a recalibrated distribution with an explicit PDF, remains computationally efficient, and has finite-sample guarantees on latent calibration (see Section 3.3).

**Latent space transformation.** LR uses the CDF $F_{\hat{L}}$ as its calibration map. We define a scalar strictly increasing transformation $r : \mathbb{R}_+ \to \mathbb{R}_+$ using the quantile function $F_{\hat{L}}^{-1}$, which maps the original latent norms $l \in \mathbb{R}_+$ to recalibrated norms as follows:

$$r(l) = F_{\hat{L}}^{-1}(F_{\rho_{\mathcal{Z}}(Z)}(l)). \tag{8}$$

We also define a vector-valued transformation $R : \mathcal{Z} \to \mathcal{Z}$ based on the scalar transformation $r$, which maps latent vectors $z$ such that $\rho_{\mathcal{Z}}(R(z)) = r(\rho_{\mathcal{Z}}(z))$. When using the Euclidean norm $\rho_{\mathcal{Z}}(z) = \|z\|$, $R$ is a radial transformation:

$$R(z) = \frac{r(\|z\|)}{\|z\|} \cdot z \quad \text{(with } R(0) = 0\text{)}. \tag{9}$$

The transformation $R$ rescales each vector $z$ along its original direction by replacing its norm $\|z\|$ with $r(\|z\|)$. This procedure defines a new latent variable $Z' = R(Z)$, and the associated recalibrated NF $\hat{T}(Z'; X)$.

**Proposition 1.** The recalibrated NF $\hat{T}(Z'; X)$ defined with the new latent variable $Z' = R(Z)$ is *latent calibrated*, i.e. $\hat{U}' = F_{\rho_{\mathcal{Z}}(Z')}(\hat{L}) \sim \mathcal{U}(0, 1)$.

*Proof.* Consider the inverse transformation $r^{-1}(l) = F_{\rho_{\mathcal{Z}}(Z)}^{-1}\left(F_{\hat{L}}(l)\right)$ for a latent norm $l \in \mathbb{R}_+$. Then, using $\rho_{\mathcal{Z}}(R(z)) = r(\rho_{\mathcal{Z}}(z))$ the following identity holds:

$$F_{\rho_{\mathcal{Z}}(Z')}(l) = F_{r(\rho_{\mathcal{Z}}(Z))}(l) = F_{\rho_{\mathcal{Z}}(Z)}\left(r^{-1}(l)\right) = F_{\hat{L}}(l), \quad \forall l \in \mathbb{R}_+. \tag{10}$$

Then, it follows that

$$\mathbb{P}(\hat{U}' \leq \alpha) = \mathbb{P}(F_{\rho_{\mathcal{Z}}(Z')}(\hat{L}) \leq \alpha) = \mathbb{P}(F_{\hat{L}}(\hat{L}) \leq \alpha) = \alpha. \tag{11}$$

$\square$

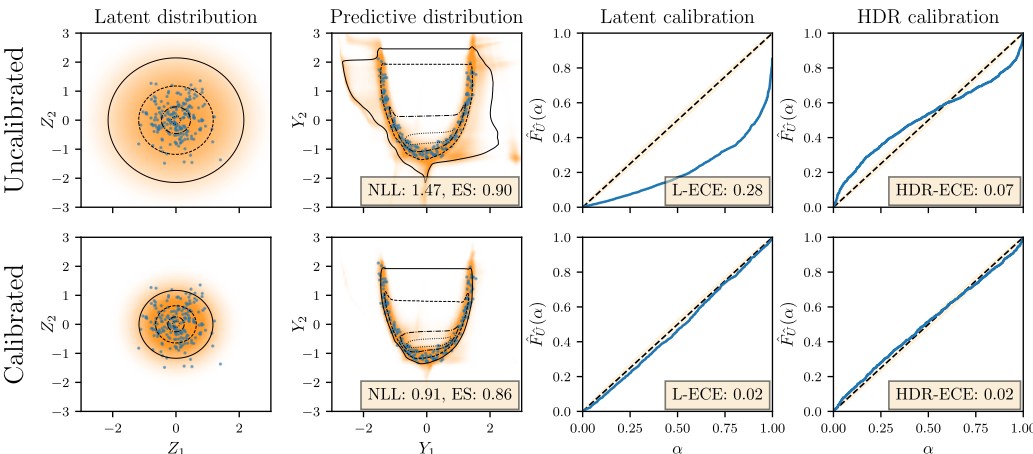

Figure 1: Illustration of LR for a bivariate output. The first column shows the latent distribution, the second column displays the predictive PDF, and the third and fourth columns show reliability diagrams for latent and HDR calibration, respectively. The first row corresponds to an uncalibrated NF, and the second is the same model after LR. Calibration points and their projections in the latent space are shown in blue. The PDF for both the latent distribution and the predictive distribution is shown in orange. Level sets of the PIT of the latent norm at levels 0.01, 0.1, 0.5, and 0.9 are indicated with black contours in the second column, and their corresponding preimages are shown in the first column. LR improves both latent calibration (third column) and HDR calibration (fourth column). Additional prediction examples on real-world datasets are presented in Section G.

**Recalibrated predictive density.** A distinctive feature of our LR recalibration procedure is that it produces a recalibrated distribution with an explicit multivariate PDF.

Note that the recalibrated NF can be interpreted as a composite transformation $\hat{T}' = \hat{T} \circ R$, applied to the original latent variable $Z$ with density $f_Z$, typically a standard multivariate Gaussian. Given $x \in \mathcal{X}$ and $y \in \mathcal{Y}$, assuming the transformation $R$ is differentiable, the recalibrated predictive density $\hat{f}'_{Y|X=x}(y)$ can be computed using the change of variables formula. Let $z' = \hat{T}^{-1}(y; x)$ and $z = R^{-1}(z')$. Then, we have

$$\hat{f}'_{Y|X=x}(y) = f_Z(z) \left|\det\left(\nabla_z R(z)\right)\right|^{-1} \left|\det\left(\nabla_y \hat{T}^{-1}(y; x)\right)\right|. \tag{12}$$

Let us consider the case where $\rho_{\mathcal{Z}}(z) = \|z\|$. The inverse transformation takes the form $R^{-1}(z') = \frac{r^{-1}(\|z'\|)}{\|z'\|} \cdot z'$ and the Jacobian determinant of $R$ can be computed efficiently as:

$$\left|\det\left(\nabla_z R(z)\right)\right| = \left(\frac{r(l)}{l}\right)^{d-1} \cdot \frac{\partial r(l)}{\partial l}, \quad \text{with } l = \|z\|. \tag{13}$$

A detailed proof is provided in Section C.1. The term $\frac{\partial r(l)}{\partial l}$ in (13) is computed using the chain rule as:

$$\frac{\partial r(l)}{\partial l} = \frac{\partial F_{\hat{L}}^{-1}(l')}{\partial l'} \cdot \frac{\partial F_{\rho_{\mathcal{Z}}(Z)}(l)}{\partial l}, \quad \text{where } l' = F_{\rho_{\mathcal{Z}}(Z)}(l). \tag{14}$$

To compute $\partial F_{\rho_{\mathcal{Z}}(Z)}(l)/\partial l$, we leverage the fact that $\rho_{\mathcal{Z}}(Z) \sim \chi_d$, whose PDF is available in closed-form and can be evaluated efficiently.

In practice, $F_{\hat{L}}$ is estimated by computing latent norms $\hat{L}_i = \ell_{\hat{T}}(X^{(i)}, Y^{(i)})$ using samples $(X^{(i)}, Y^{(i)})$ from the calibration set $\mathcal{D}_{\text{cal}}$. Section D details how this can be achieved using kernel density estimation or monotonic splines, resulting in a differentiable estimate $\hat{F}_{\hat{L}}$ of $F_{\hat{L}}$. All operations are carried out in log-space to ensure numerical stability. Figure 1 illustrates LR, with the recalibrated predictive density $\hat{f}'_{Y|X}$ shown in the second column of the second row.

### 3.3 Useful Properties of Multivariate Latent Recalibration

We present finite-sample coverage guarantees for LR and highlight its connections to conformal prediction methods. We assume that $R$ depends on an estimate $\hat{F}_{\hat{L}}$ of $F_{\hat{L}}$ based on latent norms $\hat{L}_1, \ldots, \hat{L}_n$.

**Finite-sample coverage guarantees for recalibrated latent norms.** Let us assume that the estimated calibration map $\hat{F}_{\hat{L}}$ maps the $i$-th order statistic $\hat{L}_{(i)}$ of $\hat{L}_1, \ldots, \hat{L}_n$ within a margin $\lambda/(n+1) \geq 0$ of the target quantile $i/(n+1)$, that is,

$$\hat{F}_{\hat{L}}(\hat{L}_{(i)}) \in \left[ \frac{i-\lambda}{n+1}, \frac{i+\lambda}{n+1} \right]. \tag{15}$$

Then, letting $\epsilon = \frac{1+\lambda}{n+1}$, Theorem 1 of Marx et al., 2022 yields the following finite-sample coverage guarantee for the recalibrated latent norms:

$$\mathbb{P}\left( F_{\rho_{\mathcal{Z}}(Z')}\left( \ell_{\hat{T}}(Y; X) \right) \leq \alpha \right) = \mathbb{P}\left( \hat{F}_{\hat{L}}(\hat{L}) \leq \alpha \right) \in [\alpha - \epsilon, \alpha + \epsilon], \tag{16}$$

where we used (10) for the first equality and the probabilities are taken over $X, Y$, and the recalibrated latent norms $\hat{L}_1, \ldots, \hat{L}_n$.

**Equivalence with conformal prediction sets.** We observe that the prediction sets derived from the recalibrated predictive density of LR coincide exactly with those obtained by the multivariate conformal methods CONTRA (Fang et al., 2025) and L-CP (Dheur et al., 2025). Specifically, this equivalence holds when LR uses the empirical CDF of the calibration scores $\mathcal{L} = \{\hat{L}_1, \ldots, \hat{L}_n, +\infty\}$ as its calibration map, i.e., $\hat{F}_{\hat{L}}(l) = \frac{1}{n+1} \sum_{i=1}^{n} \mathbb{1}(\hat{L}_i \leq l)$.

CONTRA and L-CP are conformal methods that construct prediction sets using the conformity score $s_{\text{CONTRA}}(x, y) = s_{\text{L-CP}}(x, y) = \ell_{\hat{T}}(y; x)$. Under this choice of calibration map, for any $x \in \mathcal{X}$ and $\alpha \in (0, 1)$, we have

$$\{y \in \mathcal{Y} : F_{\rho_{\mathcal{Z}}(Z')}(\ell_{\hat{T}}(y; x)) \leq \alpha\} = \{y \in \mathcal{Y} : \hat{F}_{\hat{L}}(\ell_{\hat{T}}(y; x)) \leq \alpha\} \tag{17}$$

$$= \{y \in \mathcal{Y} : s_{\text{CONTRA}}(x, y) \leq \hat{F}_{\hat{L}}^{-1}(\alpha)\}, \tag{18}$$

where $\hat{F}_{\hat{L}}^{-1}(\alpha) = \hat{L}_{(\lceil \alpha(n+1) \rceil)}$ denotes the $(1 - \alpha)$ right empirical quantile of the calibration scores $\mathcal{L}$. This shows that the $\alpha$-sublevel sets of the PIT of the latent norm of LR (17) correspond exactly to the conformal prediction sets produced by CONTRA and L-CP at coverage $\alpha$ (18). While this equivalence is notable, it is important to point out that the chosen calibration map $\hat{F}_{\hat{L}}$, being non-differentiable, does not yield a well-defined recalibrated predictive density function $\hat{f}'_{Y|X}$. This equivalence is summarized in the last row of Table 1.

**Equivalence of LR and QR in the single-output setting.** QR is a special case of LR where $d = 1$, $\mathcal{Z} = [0, 1]$, $Z \sim \mathcal{U}(0, 1)$, $\hat{T}^{-1}(y; x) = \hat{F}_{Y|X=x}(y)$ and $\rho_{\mathcal{Z}}(z) = z$. In this case, $R = \hat{F}_{\hat{L}}^{-1}$ and thus $\hat{T}'^{-1}(\cdot; x) = R^{-1} \circ \hat{T}^{-1}(\cdot; x) = \hat{F}_{\hat{L}} \circ \hat{F}_{Y|X=x} = \hat{F}'_{Y|X=x}$, showing that both methods perform exactly the same transformation.

# 4 Related work

Our work builds upon and contributes to generative modeling, calibration, conformal prediction, and methods that combine these concepts in the context of multi-output regression. An extended description of related works is available in Section B.

Various notions of calibration have been studied, including probabilistic (Gneiting et al., 2007), marginal (Gneiting et al., 2007) and HDR (Chung et al., 2024) calibration. Ziegel and Gneiting (2014) and Allen et al. (2024) also proposed multivariate notions of calibration but, to our knowledge, no calibration methods for these notions have been proposed.

While traditional CP focuses on univariate intervals (Romano et al., 2019; Sesia and Romano, 2021), recent multivariate CP methods create flexible regions. HPD-split (Izbicki et al., 2022) uses HPD values as scores. PCP (Wang et al., 2023) uses balls around samples. ST-DQR (Feldman et al., 2023) selects samples based on a region in a latent space and creates balls around these samples. CONTRA (Fang et al., 2025) and L-CP (Dheur et al., 2025) operate in the latent space of NFs.

Some methods explicitly merge CP and recalibration. Vovk et al. (2020) and Vovk et al. (2019) developed conformal predictive systems for calibrated univariate distributions. MCC (Marx et al., 2022) unified univariate recalibration methods under a CP lens. Our work extends this direction to multivariate outputs via a transformation in the latent space.

# 5 Experiments

We present an extensive experimental study using 29 tabular datasets widely used in prior research (Tsoumakas et al., 2011; Cevid et al., 2022; Chung et al., 2024; Feldman et al., 2023; Wang et al., 2023; Barrio et al., 2024; Camehl et al., 2024). Furthermore, while recent work on model recalibration (Chung et al., 2024; Fang et al., 2025) has primarily focused on data modalities with relatively low output dimensionality, we also include a high-dimensional output setting with an image dataset with a larger output dimension (Choi et al., 2020).

## 5.1 Datasets

**Tabular datasets.** The tabular datasets range in size from 103 to 50,000 data points, with the number of input features ($p$) varying from 1 to 368, and the number of output variables ($d$) ranging from 2 to 16. A detailed summary of these datasets is provided in Table 3 in Appendix A. Following the protocol of Chung et al. (2024), we use a 65/20/15 split for training, validation, and testing. All input features and output targets are normalized to have zero mean and unit variance on the training set. Experiments are repeated 10 times with a different random splitting. For each run, we compare the same base model with or without recalibration.

**Image dataset.** We use the AFHQ dataset (Choi et al., 2020), which consists of high-resolution animal face images. The input $x \in \mathcal{X} = \{0, 1, 2\}$ indicates one of three classes (cat, dog, or wild animal), and the output is a $256 \times 256$ RGB image $y \in \mathcal{Y} = [-1, 1]^{3 \times 256 \times 256}$, resulting in an output dimension of $d = 196{,}608$. We follow the standard split with 14,630 training instances and 1,500 test instances. To improve sample quality, Zhai et al. (2024) add Gaussian noise $\epsilon \sim \mathcal{N}(0, 0.07^2)$ to each image $y$ during training.

## 5.2 Experimental setup

**(Non-recalibrated) base model.** For the tabular datasets, we consider convex potential flows (Huang et al., 2020), masked autoregressive flows (MAFs, Papamakarios, Pavlakou, et al., 2017) and flow matching (FM, Lipman et al., 2022). A notable difference from the setup of Chung et al. (2024) is that their predictive distributions are restricted to multivariate Gaussians with diagonal covariance, whereas NFs can model dependencies between output dimensions. For the image dataset, we use the TarFlow model (Zhai et al., 2024), a transformer-based conditional NF pre-trained on AFHQ, which achieves state-of-the-art likelihood performance. As is standard, all aforementioned NFs use a latent variable $Z \sim \mathcal{N}(0, I)$. Details on these base models are provided in Section I, with hyperparameter tuning details for convex potential flows in Section E. Results for MAFs and FM are deferred to Sections K and L. In the following, we denote the non-recalibrated base model as `BASE`.

**Compared methods.** For our latent recalibration method, `LR`, we use the Euclidean norm $\rho_{\mathcal{Z}}(z) = \|z\|$ and estimate $F_L$ using kernel density estimation with a Gamma kernel; details are provided in Section D. For both tabular and image datasets, we compare `LR` with the base model `BASE`. Additionally, we include `HDR-R` for tabular datasets only, as it becomes computationally prohibitive for TarFlow. For tabular datasets, following Chung et al. (2024), the recalibration map is learned on the validation set. This avoids using additional data for calibration and ensures a fair comparison with `BASE`, but sacrifices finite-sample guarantees. For the image dataset, since no separate calibration set is available, calibration is performed on the training data. This also sacrifices finite-sample guarantees, but we will show below that it still leads to substantial improvements in calibration.

**Error metrics.** We consider several error metrics to compare the different methods. For the tabular datasets, we evaluate model calibration using the latent expected calibration error (L-ECE) and the HDR expected calibration error (HDR-ECE). Both metrics range from 0 (best) to 0.5 (worst). Predictive accuracy is assessed using two strictly proper scoring rules: negative log-likelihood (NLL) and the energy score (ES). Notably, `LR` yields a recalibrated density with a closed-form PDF, enabling direct computation of the NLL, which is not possible with `HDR-R`. Since the scales of NLL and ES vary across datasets, we report relative values, defined as the difference with the score achieved by `BASE`. All metrics are negatively oriented. Exact definitions are provided in Section E.1. For the image dataset, we report L-ECE and the bits per dimension (BPD), following Zhai et al. (2024). BPD corresponds to a rescaled version of the NLL (details in Section E.1).

### 5.3 Results

**Tabular datasets.** Figure 2 presents the normalized difference relative to BASE for NLL and ES. We observe that LR reduces the NLL on the majority of datasets. Since the NLL is a strictly proper scoring rule, this indicates that the recalibrated density $\hat{f}'_{Y|X}$ produced by LR generally provides a better fit to the true data distribution than the original model $\hat{f}_{Y|X}$. In contrast to the NLL, the ES of LR and HDR-R is largely unchanged. In Section J.3, we attribute this phenomenon to the metric's weaker dis-

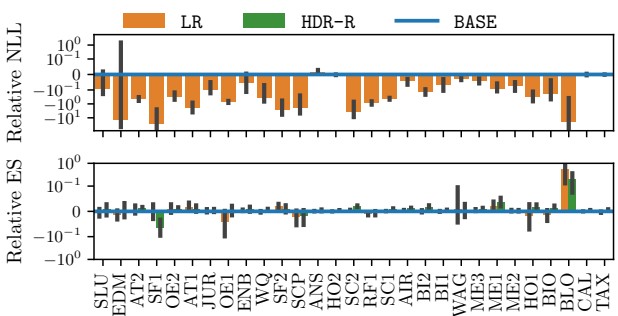

Figure 2: Relative NLL and ES on datasets sorted by size, using a convex potential flow model.

criminative ability relative to misspecifications in variance, correlation, and overall dependency structure. A notable exception is the increased ES on the blog_data (BLO) dataset, where one output is discrete and a single value is repeated across 64% of the instances. Based on this observation, we suggest not using LR on datasets with discrete outputs.

Figure 3 also shows the L-ECE (as a measure of latent calibration) and HDR-ECE (as a measure of HDR calibration), respectively. We see that BASE exhibits significant latent miscalibration across many datasets, with L-ECE values reaching up to 0.25 out of a maximum of 0.5. In contrast, LR consistently and substantially reduces L-ECE, demonstrating its effectiveness in achieving the desired latent calibration. Moreover, L-ECE tends to decrease as dataset size increases, which aligns with the finite-sample guarantees discussed in Section 3.3. Reliability diagrams in Section H further confirm this improvement. Additional experiments on a misspecified model are provided in Section M, with significant improvements given by LR across all metrics including ES.

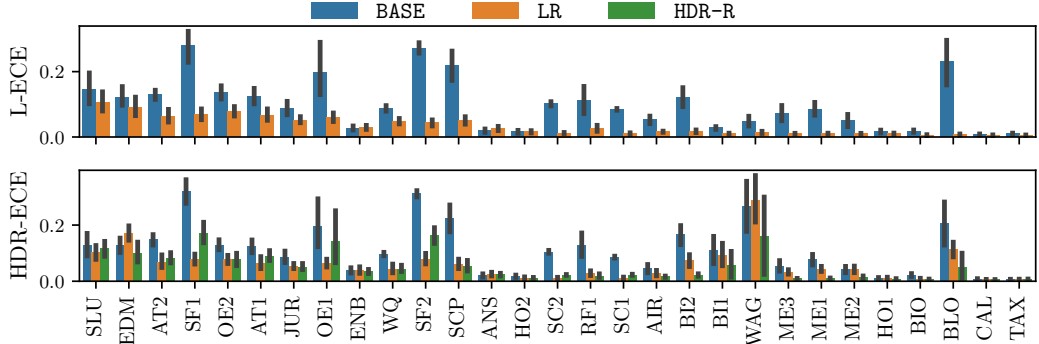

Figure 3: L-ECE and HDR-ECE on datasets sorted by size, using a convex potential flow model.

In Figure 3, while LR does not explicitly target HDR calibration, we observe that it significantly improves HDR-ECE compared to BASE on most datasets, often performing on par with HDR-R. As expected, HDR-R achieves low HDR-ECE values by design. These results suggest that improving latent calibration also enhances the calibration of HDRs.

Access to a full, calibrated PDF is crucial for any task requiring estimation of the probability mass within an *arbitrary, non-standard region* of the output space, a capability that set-based methods (like CP) or pure sampling-based methods (like HDR-R) do not provide. A direct application is anomaly detection, where low-density points are classified as anomaly (Rozner et al., 2024; Perini et al., 2024). Other examples include risk assessment in engineering, targeted material design, or optimal control. To make the benefits of a full PDF concrete, we provide an experiment on a decision-making task in Section F.

Further detailed results with the same convex potential flow base predictor are provided in Section J. Section J.1 shows that the primary NLL gain obtained by LR is due to finding more "plausible" latent codes for the observed data under the base latent distribution. Section J.2 shows that LR significantly

Table 2: Performance of LR compared to BASE on the AFHQ dataset with TarFlow (standard errors across 20 evaluations).

| | L-ECE | | BPD | |
|---|---|---|---|---|
| BASE | LR | BASE | LR |
| $0.474_{0.000625}$ | $0.00895_{0.00160}$ | $5.477_{3.523e\text{-}05}$ | $5.465_{1.772e\text{-}05}$ |

improves the time to compute the calibration map compared to HDR-R. Section J.3 hypothesizes through theoretical and empirical considerations that the reason the ES remains largely unchanged after LR is due to its relative insensitivity to misspecifications in variance and correlation.

**Image dataset.** The goal of our image data experiment is to understand the behaviour of latent calibration and recalibration with high-dimensional outputs. Table 2 shows that BASE suffers from severe latent miscalibration, with L-ECE values approaching the maximum of 0.5. LR dramatically improves latent calibration, reducing L-ECE to below 0.01. We also report the bits-per-dimension (BPD), a scaled version of the NLL. Notably, LR does not degrade the original NLL; in fact, it slightly reduces it. LR preserves the visual quality of the samples from the base model, with no perceptually visible changes, which aligns with the very small change we observed in NLL.

## 6 Conclusion and limitations

We introduced latent recalibration (LR), a novel post-hoc method for calibrating conditional normalizing flows in multi-output regression. By transforming the latent space based on calibration scores derived from latent distances, LR achieves latent calibration, ensuring that prediction regions defined in the latent space have correct coverage. Unlike many conformal prediction methods that only output sets, and unlike sampling-based recalibration methods, LR yields a fully specified, recalibrated PDF. This offers significant advantages in terms of computational efficiency and applicability to tasks requiring density estimates. Our extensive experiments on tabular and high-dimensional image data demonstrate that LR consistently improves NLL, latent calibration, and HDR calibration.

We identify the main limitations of LR as follows. Firstly, LR intentionally adjusts only the magnitude of latent vectors, not their direction, and thus cannot fix miscalibration arising from errors in the orientation of the learned latent manifold. While LR can only perform simple adjustments, this allows simplifying the difficult multivariate calibration problem into a tractable univariate one (calibrating norms). This enables connections with conformal prediction and recalibration methods, and has good empirical performance. Secondly, LR requires the norm of the latent distribution to follow a simple distribution. This is usually the case, as normalizing flows predominantly use a standard Gaussian latent variable. Thirdly, LR requires an invertible transformation between the response and latent spaces, and a latent random variable with a known, tractable density, which makes it incompatible with models such as variational auto-encoders (Kingma and Welling, 2014) or denoising diffusion probabilistic models (Ho et al., 2020). Instead, NF and FM models are natural fits for LR. Despite these considerations, LR provides a practical and effective tool for obtaining reliable, calibrated multivariate predictive distributions from generative models.

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

# A Datasets

Table 3 shows the datasets considered in our experiments, with associated reference papers. The datasets are characterized by their total number of instances, number of features $p$ and number of outcomes $d$. The preprocessing follows the setup described in Grinsztajn et al., 2022.

Table 3: Lists of evaluated tabular datasets.

| Paper | Dataset | Abbreviation | Total size | $p$ | $d$ |
|---|---|---|---|---|---|
| Tsoumakas et al., 2011 | slump | SLU | 103 | 7 | 3 |
| | edm | EDM | 154 | 16 | 2 |
| | atp7d | AT2 | 296 | 355 | 6 |
| | sf1 | SF1 | 323 | 31 | 3 |
| | oes97 | OE2 | 334 | 263 | 16 |
| | atp1d | AT1 | 337 | 354 | 6 |
| | jura | JUR | 359 | 15 | 3 |
| | oes10 | OE1 | 403 | 298 | 16 |
| | enb | ENB | 768 | 3 | 2 |
| | wq | WQ | 1060 | 16 | 14 |
| | sf2 | SF2 | 1066 | 31 | 3 |
| | scpf | SCP | 1137 | 8 | 3 |
| Barrio et al., 2024 | ansur2 | ANS | 1986 | 1 | 2 |
| Camehl et al., 2024 | households | HO2 | 7207 | 14 | 4 |
| Tsoumakas et al., 2011 | scm20d | SC2 | 8966 | 60 | 16 |
| | rf1 | RF1 | 9005 | 64 | 8 |
| | scm1d | SC1 | 9803 | 279 | 16 |
| Cevid et al., 2022 | births2 | BI2 | 10000 | 24 | 4 |
| | air | AIR | 10000 | 15 | 6 |
| | births1 | BI1 | 10000 | 23 | 2 |
| | wage | WAG | 10000 | 78 | 2 |
| Feldman et al., 2023 | meps_21 | ME3 | 15656 | 138 | 2 |
| | meps_19 | ME1 | 15785 | 138 | 2 |
| | meps_20 | ME2 | 17541 | 138 | 2 |
| | house | HO1 | 21613 | 17 | 2 |
| | bio | BIO | 45730 | 8 | 2 |
| | blog_data | BLO | 50000 | 269 | 2 |
| Barrio et al., 2024 | calcofi | CAL | 50000 | 1 | 2 |
| Wang et al., 2023 | taxi | TAX | 50000 | 4 | 2 |

# B Extended related work

**Normalizing flows.** Various NF architectures exist, including RealNVP (Dinh et al., 2016), MAF (Papamakarios, Pavlakou, et al., 2017), Glow (Kingma and Dhariwal, 2018), spline flows (Durkan et al., 2019), convex potential flows (Huang et al., 2020) and transformer flows (Zhai et al., 2024). NFs are well-suited for LR due to their invertible mapping and explicit density. Radial flows (Rezende and Mohamed, 2015) are a special case of (9) with $r(t) = t \cdot (\alpha + t + \beta)/(\alpha + t)$ where $\alpha \in \mathbb{R}_+$ and $\beta \in \mathbb{R}$ are learned parameters.

**Conditional notions of calibration.** Multiple extensions of notions of calibration have been proposed by imposing different conditions, including group calibration (Pleiss et al., 2017), distribution calibration (or auto-calibration) (Song et al., 2019; Tsyplakov, 2013), individual calibration (Zhao et al., 2020), and threshold calibration (Sahoo et al., 2021). Latent calibration could be extended by imposing similar conditions.

**Calibration methods.** Improving calibration often involves post-hoc recalibration or regularization during training. *Recalibration* methods adjust pre-trained models. Notable methods include Kuleshov

et al., 2018; Kuleshov and Deshpande, 2022; Chung et al., 2024. *Regularization* methods incorporate calibration objectives into training (Utpala and Rai, 2020; Marx et al., 2023; Dheur and Ben Taieb, 2023). However, these methods target univariate settings and can trade off predictive accuracy (NLL, CRPS) for calibration (Yoon et al., 2023; Dheur and Ben Taieb, 2023). Dheur and Ben Taieb (2024) integrated QR end-to-end into training, followed by post-hoc recalibration.

## C  Proofs

### C.1  Jacobian determinant of the radial transform

Recall that the transformation $R$ is defined as

$$R(z) = \frac{r(l)}{l} z \tag{19}$$

where $l = \|z\|$, for $z \neq 0$. It maps $z$ to a new vector $R(z)$ such that its norm becomes $r(l)$ while its direction $z/l$ is preserved (for $z \neq 0$). We analyze this transformation using hyperspherical coordinates $(l, \omega_1, \ldots, \omega_{d-1})$, where $l = \|z\|$ is the radial distance and $(\omega_1, \ldots, \omega_{d-1})$ are the angular coordinates. The transformation $R$ maps these coordinates from $(l, \omega_1, \ldots, \omega_{d-1})$ to $(r(l), \omega_1, \ldots, \omega_{d-1})$, as only the radial distance is altered.

The Cartesian volume element $\mathrm{d}^d z$ is related to the hyperspherical volume element by $\mathrm{d}^d z = l^{d-1} \mathrm{d}l \, \mathrm{d}\Omega_{d-1}$, where $\mathrm{d}\Omega_{d-1}$ is the surface element on the unit $(d-1)$-sphere. Under the transformation $R$, the new radial coordinate is $l' = r(l)$, so its differential is $\mathrm{d}l' = \frac{\partial r(l)}{\partial l} \mathrm{d}l$. The angular part $\mathrm{d}\Omega_{d-1}$ remains unchanged. The transformed volume element $\mathrm{d}^d R(z)$ is thus given by:

$$\mathrm{d}^d R(z) = (r(l))^{d-1} \left( \frac{\partial r(l)}{\partial l} \mathrm{d}l \right) \mathrm{d}\Omega_{d-1}. \tag{20}$$

The Jacobian determinant $|\det(\nabla_z R(z))|$ is the ratio of the transformed volume element $\mathrm{d}^d R(z)$ to the original volume element $\mathrm{d}^d z$:

$$|\det(\nabla_z R(z))| = \frac{(r(l))^{d-1} \frac{\partial r(l)}{\partial l} \mathrm{d}l \, \mathrm{d}\Omega_{d-1}}{l^{d-1} \mathrm{d}l \, \mathrm{d}\Omega_{d-1}} = \left( \frac{r(l)}{l} \right)^{d-1} \frac{\partial r(l)}{\partial l}, \tag{21}$$

which corresponds to (13). This holds for $l = \|z\| > 0$. Since $r : \mathbb{R}_+ \to \mathbb{R}_+$, $r(l) \geq 0$. Furthermore, $r(l)$, as defined by (8), is a composition of non-decreasing functions (a CDF and an inverse CDF), making it non-decreasing, so $\frac{\partial r(l)}{\partial l} \geq 0$. Thus, the expression is inherently non-negative.

## D  Differentiable calibration maps using density estimation

To obtain a differentiable calibration map $\hat{F}_{\hat{L}}$, we estimate the PDF $\hat{f}_{\hat{L}}$ of the calibration data using density estimation. We identified two approaches that performed well in our experiments.

As an implementation detail for both approaches, density estimation was generally improved by first applying the transformation $g(t) = t^{1/3}$ to the calibration scores $\{\hat{L}_i\}_{i=1}^n$. After density estimation in the transformed space, the data is rescaled using the inverse transformation $g^{-1}(t) = t^3$.

### D.1  Kernel density estimation with Gamma kernels

We found that kernel density estimation (KDE) with Gamma kernels is effective because the Gamma distribution has positive support, which is appropriate for the calibration scores $\hat{L}_i \geq 0$. Let $\Gamma(\zeta, \lambda)$ denote a Gamma distribution with shape $\zeta > 0$ and rate $\lambda > 0$. A Gamma distribution $\Gamma(\mu\lambda, \lambda)$ has a mean of $\mu$. We center a Gamma kernel at each calibration score $\hat{L}_i$, using the distribution $\Gamma(\hat{L}_i\lambda, \lambda)$, which has a mean of $\hat{L}_i$.

The resulting estimated CDF $\hat{F}_{\hat{L}}(t)$ is given by the average of the individual kernel CDFs:

$$\hat{F}_{\hat{L}}(t) = \frac{1}{n} \sum_{i=1}^n F_{\Gamma(\hat{L}_i\lambda, \lambda)}(t). \tag{22}$$

The rate parameter $\lambda$ is chosen by minimizing the NLL of the calibration dataset under the KDE model. This is done using 10-fold cross-validation over the grid $\left\{10^{-5+10\cdot\frac{i}{99}}\right\}_{i=0}^{99}$. This hyperparameter selection process is efficient and performed once per run.

Figure 4 shows an example fit on all datasets, illustrating the empirical and estimated smooth CDFs (left $y$ axis) and the estimated log PDF (right $y$ axis).

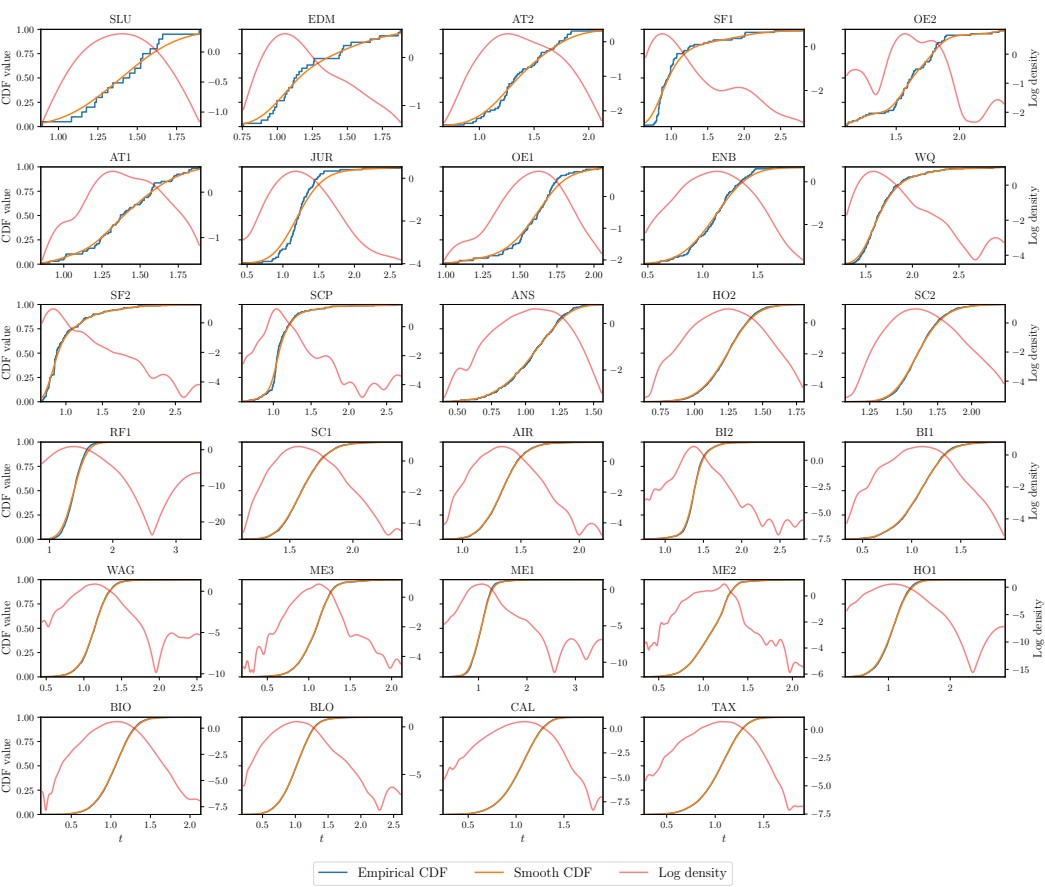

Figure 4: Density estimation using KDE with a Gamma kernel.

## D.2 Rational Quadratic Splines

Rational Quadratic Splines (Durkan et al., 2019) provide a flexible framework for defining invertible and differentiable transformations. A base spline $\Phi$ maps $[-1, 1]$ to $[-1, 1]$. To handle the unbounded domain of the latent norms, we use the transformation $\Psi = \tanh^{-1} \circ \Phi \circ \tanh$, which maps $\mathbb{R}$ to $\mathbb{R}$ and retains invertibility and differentiability. This transformation is used to model the distribution of the latent norms by learning a mapping from a standard Gaussian distribution to the data distribution.

For training, the data is normalized to have zero mean and unit variance. The parameters of the spline are optimized to minimize the NLL of the calibration dataset under the defined model. Optimization is performed using Adam (Kingma and Ba, 2014). To maximize data information, we perform early stopping on the training dataset itself and stop if the loss did not improve by 1e-4 for 50 epochs. Overfitting is prevented by limiting the number of bins and thus the flexibility of the spline. Specifically, we use 4 bins if $n \leq 30$, 5 bins if $n \leq 50$, 6 bins if $n \leq 70$, 7 bins if $n \leq 80$, 8 bins if $n \leq 90$, and 9 bins if $n \leq 100$.

Similar to Figure 4, Figure 5 shows an example fit for all datasets.

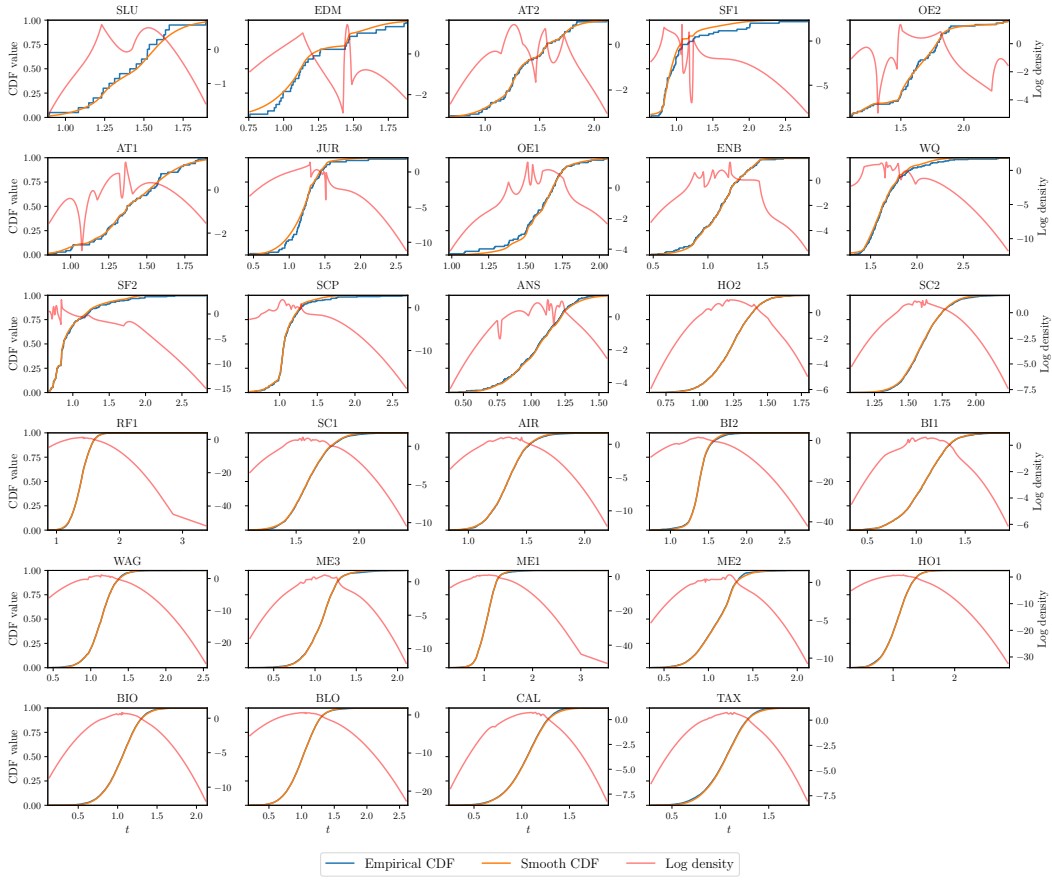

Figure 5: Density estimation using a rational quadratic spline.

### D.3 Challenges with numerical precision

In this section, we address a potential alternative approach and explain why it is impractical due to numerical precision constraints. Theoretically, one could attempt to estimate the calibration map $\hat{F}_{\hat{U}}$ using density estimation on the quantity $\hat{U} = F_{\rho_{\mathcal{Z}}(Z)}(\ell_{\hat{T}}(Y;X))$. Since $\hat{U}$ is expected to follow a standard uniform distribution under ideal conditions, estimating its density might appear practical.

However, this approach faces significant numerical precision issues, particularly when the latent space dimensionality $d$ is large. When $d$ is large, the CDF $F_{\rho_{\mathcal{Z}}(Z)}$ for $\rho_{\mathcal{Z}}(Z) \sim \chi_d$ becomes extremely steep around its mode $\sqrt{d-1}$. For example, in our image application with $d = 196,608$, a proportion $99\%$ of the probability mass is concentrated in the narrow interval $[440.7, 446.2]$. In single-precision floating-point arithmetic, the CDF saturates quickly: $F_{\chi_d}(t)$ is numerically $0.0$ for $t < 433.4$ and $1.0$ for $t > 447.2$.

If the latent model is miscalibrated, the values of $\ell_{\hat{T}}(Y;X)$ for the calibration data can fall outside this narrow range where the CDF has fine-grained variation. This results in many $\hat{U}$ values being numerically $0.0$ or $1.0$. For smaller dimensions, similar issues can occur, although less frequently. For instance, with $d = 1$, $F_{\chi_1}(t)$ is numerically $1.0$ for $t > 5.54$ in single precision. When a significant portion of the calibration data for $\hat{U}$ consists of values numerically identical to $0.0$ or $1.0$, accurate density estimation becomes impossible.

For this reason, in Section 3.2, we based our calibration map on density estimation of $\hat{L} = \ell_{\hat{T}}(Y;X)$ directly, using $\hat{F}_{\hat{L}}$.

# E  Additional details on experimental setup

Computing the main tabular data results requires approximately 24 hours on an RTX A6000 GPU, and reproducing the image results requires approximately 6 hours on an RTX 6000 GPU. Experiments can require up to 48 GB of VRAM, primarily due to large batch sizes during sampling for evaluation metrics. Decreasing the batch size reduces VRAM requirements but increases computation time.

For tabular datasets, we tune hyperparameters using grid search, selecting those that yield the lowest NLL on the validation set. For the convex potential flow, the number of units in the input convex neural network is chosen from $[10, 20, 40]$, the number of layers from $[2, 3, 5]$, and the learning rate from $[5 \times 10^{-3}, 10^{-3}, 2 \times 10^{-4}]$. All models are trained by minimizing the NLL with the Adam optimizer (Kingma and Ba, 2014) using a batch size of 1024.

## E.1  Evaluation metrics

**NLL.**  We compute the average NLL over the test set as $\mathcal{D}_{\text{test}}$:

$$\widehat{\text{NLL}} = \frac{1}{|\mathcal{D}_{\text{test}}|} \sum_{(X,Y) \in \mathcal{D}_{\text{test}}} -\log \hat{f}_{Y|X=X}(Y). \tag{23}$$

**L-ECE.**  For each test point $(X^{(i)}, Y^{(i)}) \in \mathcal{D}_{\text{test}}$, we compute the PIT of the latent norm $\hat{U}_i = F_{\rho_{\mathcal{Z}}(Z)}\left(\ell_{\hat{T}}(Y^{(i)}; X^{(i)})\right)$. The Latent Expected Calibration Error (L-ECE) is then estimated as the $L_1$ distance between the empirical CDF of $\{\hat{U}_i\}_{i=1}^{|\mathcal{D}_{\text{test}}|}$ and the uniform CDF:

$$\widehat{\text{L-ECE}} = \frac{1}{|\mathcal{D}_{\text{test}}|} \sum_{j=1}^{|\mathcal{D}_{\text{test}}|} \left| \hat{U}_{(j)} - \frac{j}{|\mathcal{D}_{\text{test}}| + 1} \right|, \tag{24}$$

where $\hat{U}_{(j)}$ denotes the $j$-th order statistic of the computed PIT values.

**Energy Score.**  For each test point $(X, Y) \in \mathcal{D}_{\text{test}}$, we generate two independent sets of $K$ samples, $\mathcal{S}_x$ and $\mathcal{S}'_x$, from the predictive distribution $\hat{f}_{Y|X=x}(\cdot)$. The Energy Score (ES) is estimated as:

$$\widehat{\text{ES}} = \frac{1}{|\mathcal{D}_{\text{test}}|} \sum_{(X,Y) \in \mathcal{D}_{\text{test}}} \left( \frac{1}{K} \sum_{\hat{y} \in \mathcal{S}_x} \|\hat{y} - Y\| - \frac{1}{2K^2} \sum_{\hat{y} \in \mathcal{S}_x, \hat{y}' \in \mathcal{S}'_x} \|\hat{y} - \hat{y}'\| \right). \tag{25}$$

In our experiments, we use $K = 100$.

**HDR-ECE.**  For each test point $(X^{(i)}, Y^{(i)}) \in \mathcal{D}_{\text{test}}$, we compute $G_i = \text{HPD}_{\hat{f}_{Y|X=X^{(i)}}}(Y^{(i)})$, as defined in Table 1. The HDR Expected Calibration Error (HDR-ECE) is estimated similarly to L-ECE:

$$\widehat{\text{HDR-ECE}} = \frac{1}{|\mathcal{D}_{\text{test}}|} \sum_{j=1}^{|\mathcal{D}_{\text{test}}|} \left| G_{(j)} - \frac{j}{|\mathcal{D}_{\text{test}}| + 1} \right|, \tag{26}$$

where $G_{(j)}$ is the $j$-th order statistic of the computed HDR pre-ranks. Note that computing the HDR-ECE for `HDR-R` exactly is not possible as `HDR-R` does not yield an explicit recalibrated density $\hat{f}'_{Y|X}$. Following Chung et al. (2024), we use the density $\hat{f}_{Y|X}$ of the original (non-recalibrated) model for `HDR-R` when evaluating its HDR-ECE.

**BPD.**  For image datasets, we report the Bits Per Dimension (BPD), calculated as in Zhai et al. (2024). where $d$ is the output dimensionality (e.g., $d = 3 \times 256 \times 256$ for AFHQ). The BPD is then:

$$\widehat{\text{BPD}} = (\widehat{\text{NLL}}/d + \log 128)/\log 2. \tag{27}$$

Here, the $\log 128$ term accounts for the scaling of pixel values from $[0, 255]$ to $[-1, 1]$, and division by $\log 2$ converts the NLL from nats to bits.

**Algorithm 1** Pre-rank recalibration.

---

1: **Input:** Calibration dataset $\mathcal{D}_{\text{cal}}$, pre-rank $g : \mathcal{X} \times \mathcal{Y} \to \mathbb{R}$, number of samples $K$, number of bins $B$, base predictor with predictive distribution $\hat{P}_{Y|X}$, test input $x_{\text{test}}$.
2: **Calibration:**
3: **for** $(X^{(i)}, Y^{(i)}) \in \mathcal{D}_{\text{cal}}$
4:    **for** $k = 1$ to $K$
5:      $\hat{Y}^{(k,i)} \sim \hat{P}_{Y|X=X^{(i)}}$
6:      $\hat{G}^{(k,i)} \leftarrow g\left(X^{(i)}, \hat{Y}^{(k,i)}\right)$
7:    Define $\hat{F}_{\hat{G}|X=X^{(i)}}(c) = \frac{1}{K} \sum_{k=1}^{K} \mathbb{1}\left(\hat{G}^{(k,i)} \leq c\right)$
8:    $\hat{U}_i \leftarrow \hat{F}_{\hat{G}|X=X^{(i)}}\left(g\left(X^{(i)}, Y^{(i)}\right)\right)$
9: Define $\hat{F}_{\hat{U}}(u) = \frac{1}{|\mathcal{D}_{\text{cal}}|} \sum_{i=1}^{|\mathcal{D}_{\text{cal}}|} \mathbb{1}\left(\hat{U}_i \leq u\right)$ // Calibration map
10: **Prediction:**
11: **for** $k = 1$ to $K$
12:    $\hat{Y}^{(k)} \sim \hat{P}_{Y|X=x_{\text{test}}}$
13:    $\hat{G}^{(k)} \leftarrow g\left(x_{\text{test}}, \hat{Y}^{(k)}\right)$
14: Define a permutation $\pi$ such that $\hat{G}^{(\pi(1))} \leq \cdots \leq \hat{G}^{(\pi(K))}$
15: $\mathcal{S}' \leftarrow \emptyset$ // Initial set of samples
16: **for** $b = 1$ to $B$
17:    $n_b \leftarrow \lfloor K\hat{F}_{\hat{U}}(\frac{b}{B})\rfloor - \lfloor K\hat{F}_{\hat{U}}(\frac{b-1}{B})\rfloor$ // Number of resamples
18:    **if** $n_b > 0$
19:      $\mathcal{B}_b \leftarrow \left\{\left\lfloor \frac{K(b-1)}{B}\right\rfloor + 1, \ldots, \left\lfloor \frac{Kb}{B}\right\rfloor\right\}$
20:      $\mathcal{S}_b \leftarrow \{\hat{Y}^{(\pi(k))}\}_{k \in \mathcal{B}_b}$ // Samples pool
21:      $\{\tilde{Y}^{(k)}\}_{k=1}^{n_b} \sim P_{\mathcal{S}_b}$ // Resampling with replacement
22:      $\mathcal{S}' \leftarrow \mathcal{S}' \cup \{\tilde{Y}^{(k)}\}_{k=1}^{n_b}$
23: **return** recalibrated predictive samples $\mathcal{S}'$

---

**Relative NLL or ES.** To better visualize improvements in NLL or ES relative to the baseline model BASE, we report the difference in these scores, normalized by the absolute value of the score of BASE. For example, the relative NLL for LR is computed as $(\widehat{\text{NLL}}_{\text{LR}} - \widehat{\text{NLL}}_{\text{BASE}})/|\widehat{\text{NLL}}_{\text{BASE}}|$. A negative value indicates improvement by LR.

### E.2 HDR recalibration

For completeness, Algorithm 1 provides the exact recalibration procedure of the HDR-R baseline (Chung et al., 2024) introduced in Section 2.3. Instead of the HDR recalibration algorithm in Chung et al., 2024, we present a direct generalization to any pre-rank $g : \mathcal{X} \times \mathcal{Y} \to \mathbb{R}$, which we call pre-rank recalibration. HDR-R is a special case when $g(x, y) = -\hat{f}_{Y|X=x}(y)$.

## F   Decision-making experiment

To make the benefits of a full PDF concrete, we have conducted an experiment on a decision-making task.

**Experiment setup.** We use the SLUMP dataset, where inputs are ingredients for producing concrete and outputs $Y = (S, F, C) \in \mathbb{R}^3$ are three concrete properties. A manufacturer must decide among 3 actions $\mathcal{A} = \{A, B, D\}$ whether a given batch of ingredients is suitable for one of two projects ($A$ or $B$), each with specific requirement regions, or if it should be discarded ($D$). The decision has different financial utilities and risks:

- Requirements for Project A: $7 \leq S \leq 20, 55 \leq F \leq 65, 25 \leq C \leq 40$.
- Requirements for Project B: $20 \leq S \leq 29, 70 \leq F \leq 100, 15 \leq C \leq 30$.

Table 4: Comparison of estimation strategies and methods. The best performing combination is highlighted in bold.

| Method | Estimation Strategy | Average Utility |
|--------|---------------------|-----------------|
| BASE | Sampling | $62.53 \pm 11.33$ |
| HDR-R | Sampling | $32.38 \pm 10.81$ |
| BASE | PDF (Numerical Integration) | $76.23 \pm 11.99$ |
| LR | **PDF (Numerical Integration)** | **$113.31 \pm 12.91$** |

The expected utility for an agent with policy $a : \mathcal{X} \to \mathcal{A}$ is given by

$$\mathbb{E}[u(Y, a(X))] \text{ with } u(y, a) = \begin{cases} 2000, & \text{if } a = A \text{ and } y \in \text{Region}_A \\ -30, & \text{if } a = A \text{ and } y \notin \text{Region}_A \\ 1500, & \text{if } a = B \text{ and } y \in \text{Region}_B \\ -15, & \text{if } a = B \text{ and } y \notin \text{Region}_B \\ -10, & \text{if } a = C. \end{cases} \tag{28}$$

The optimal action is chosen by maximizing the estimated expected utility. This requires estimating the probabilities $\hat{P}(Y \in \text{Region}_A \mid X)$ and $\hat{P}(Y \in \text{Region}_B \mid X)$, which are computed using two approaches: (1) Monte Carlo estimation with 125 samples, or (2) numerical integration of the PDF over a 5x5x5 grid via the trapezoidal rule.

The agent acts according to the policy $a^*(X) = \arg\max_{a \in \{A,B,C\}} u_a(X)$ with

$$u_A(X) = 2000\hat{P}(Y \in \text{Region}_A \mid X) - 30\hat{P}(Y \notin \text{Region}_A \mid X)$$
$$u_B(X) = 1500\hat{P}(Y \in \text{Region}_B \mid X) - 15\hat{P}(Y \notin \text{Region}_B \mid X)$$
$$u_C(X) = -10.$$

**Results.** Table 4 show two key observations:

1. Using the PDF via numerical integration leads to better decisions (higher utility) than relying on a finite number of samples.

2. The improved calibration from LR provides a more accurate PDF, leading to a significant further increase in utility. HDR-R, which relies on resampling from the original uncalibrated density, actually harmed decision quality in this task.

This demonstrates a concrete scenario where an explicit, calibrated PDF is not just a theoretical advantage but a practical necessity for optimal decision-making.

# G  Examples of predictive distributions on real-world tabular datasets

Figures 6 and 7 display examples of predictive PDFs on real-world tabular datasets with two-dimensional outputs ($d = 2$). Each row corresponds to a different dataset. For each dataset, two random test instances, $(x^{(1)}, y^{(1)})$ and $(x^{(2)}, y^{(2)})$ from $\mathcal{D}_{\text{test}}$, are shown.

Columns 1 and 3 show the predictive densities from the uncalibrated base predictor BASE (i.e., $\hat{f}_{Y|X=x^{(1)}}(\cdot)$ and $\hat{f}_{Y|X=x^{(2)}}(\cdot)$). Columns 2 and 4 show the corresponding predictive densities from the LR-recalibrated model (i.e., $\hat{f}'_{Y|X=x^{(1)}}(\cdot)$ and $\hat{f}'_{Y|X=x^{(2)}}(\cdot)$). All densities are visualized in orange. The true target observations ($y^{(1)}$ and $y^{(2)}$) are marked with a blue dot. The negative log-likelihood of the true target under the respective predictive density is provided in the bottom right corner of each plot. Black contour lines indicate level sets of the PIT of the latent norm ($F_{\rho_{\mathcal{Z}}(Z')}(\ell_{\hat{T}}(y; x))$ for the LR model, and $F_{\rho_{\mathcal{Z}}(Z)}(\ell_{\hat{T}}(y; x))$ for the BASE model) at probability levels 0.01, 0.1, 0.5, and 0.9.

In many cases, when the BASE model is already reasonably well-calibrated, LR applies a subtle adjustment that is difficult to perceive visually. In other instances, the recalibration effect is more pronounced, visibly altering the shape and spread of the predictive distribution to better align with

latent calibration. Note that two-dimensional datasets often benefit from smaller NLL improvements according to Table 10, suggesting that stronger adjustments should be perceived in higher dimensions.

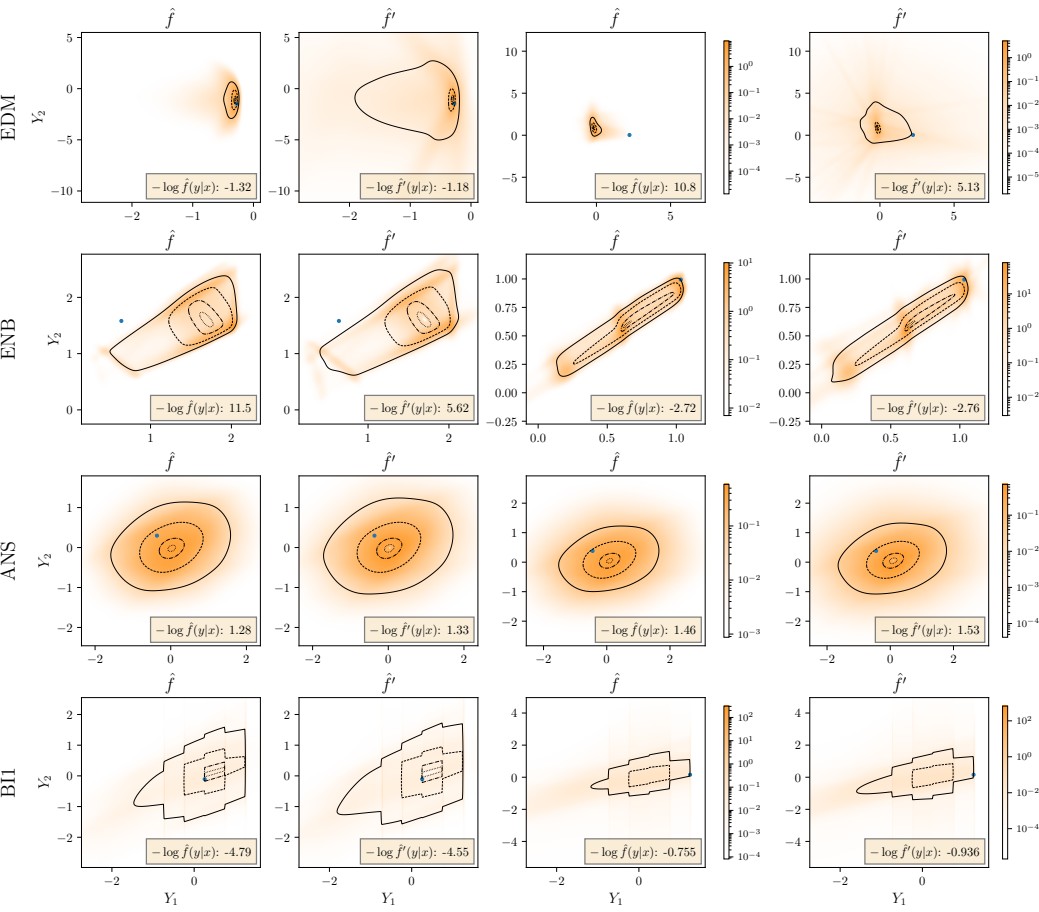

Figure 6: Examples of 2D predictive densities on real-world datasets for random test points $(x, y) \in \mathcal{D}_{\text{test}}$.

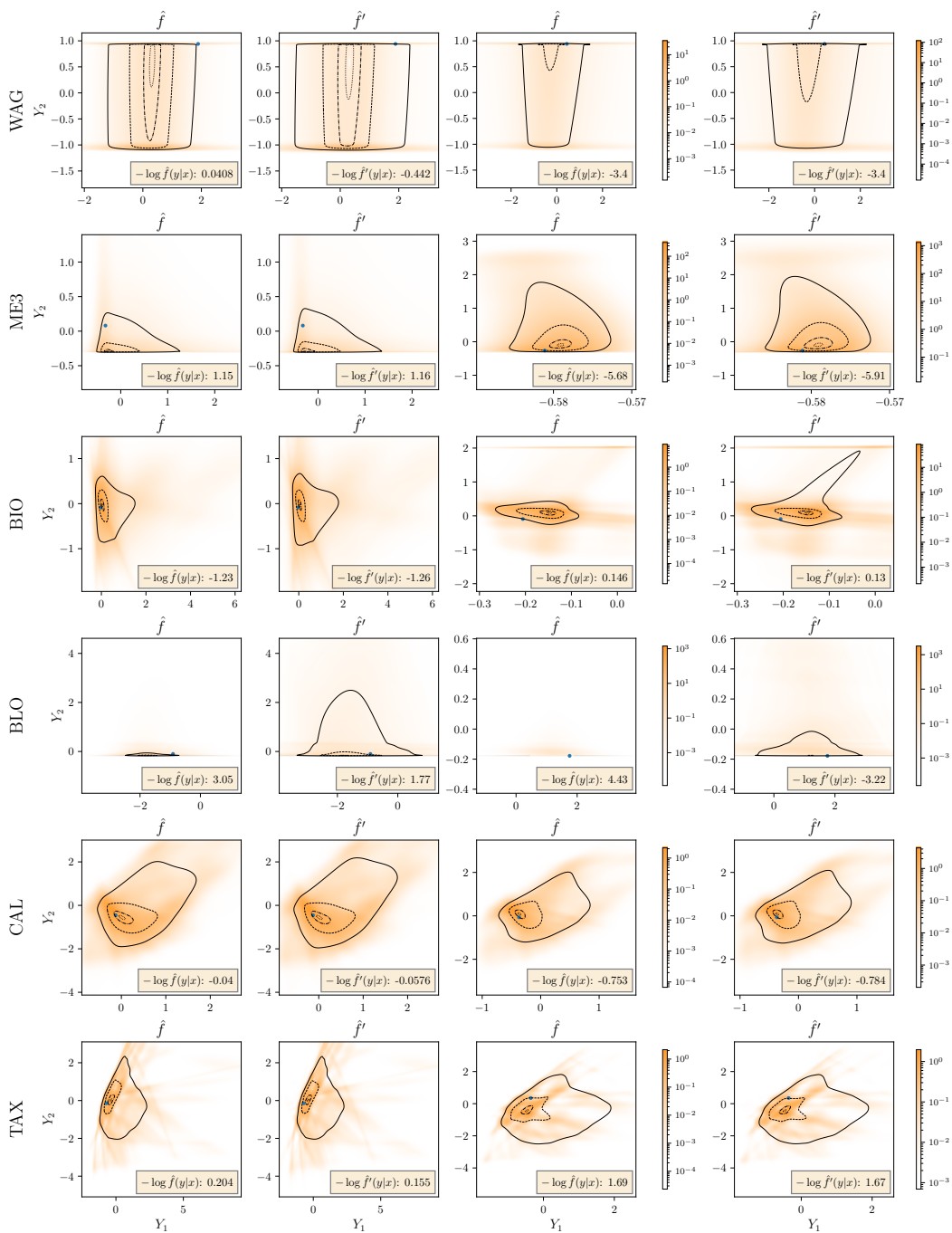

Figure 7: Examples of 2D predictive densities on real-world datasets for random test points $(x, y) \in \mathcal{D}_{\text{test}}$.

# H  Reliability diagrams

Figure 8 shows reliability diagrams for latent calibration. These diagrams plot the nominal probability levels $\alpha \in [0, 1]$ against the empirical probabilities $\hat{F}_{\hat{U}}(\alpha)$, where $\hat{U} = F_{\rho_{\mathcal{Z}}(Z)}\left(\ell_{\hat{T}}(Y; X)\right)$ are the PIT values computed on the test set. We also report 90% consistency bands, represented by the shaded area around the diagonal, as described by Gneiting et al. (2023). The BASE model often exhibits miscalibration (deviations from the diagonal), whereas LR consistently aligns closely with the diagonal, demonstrating significantly improved latent calibration.

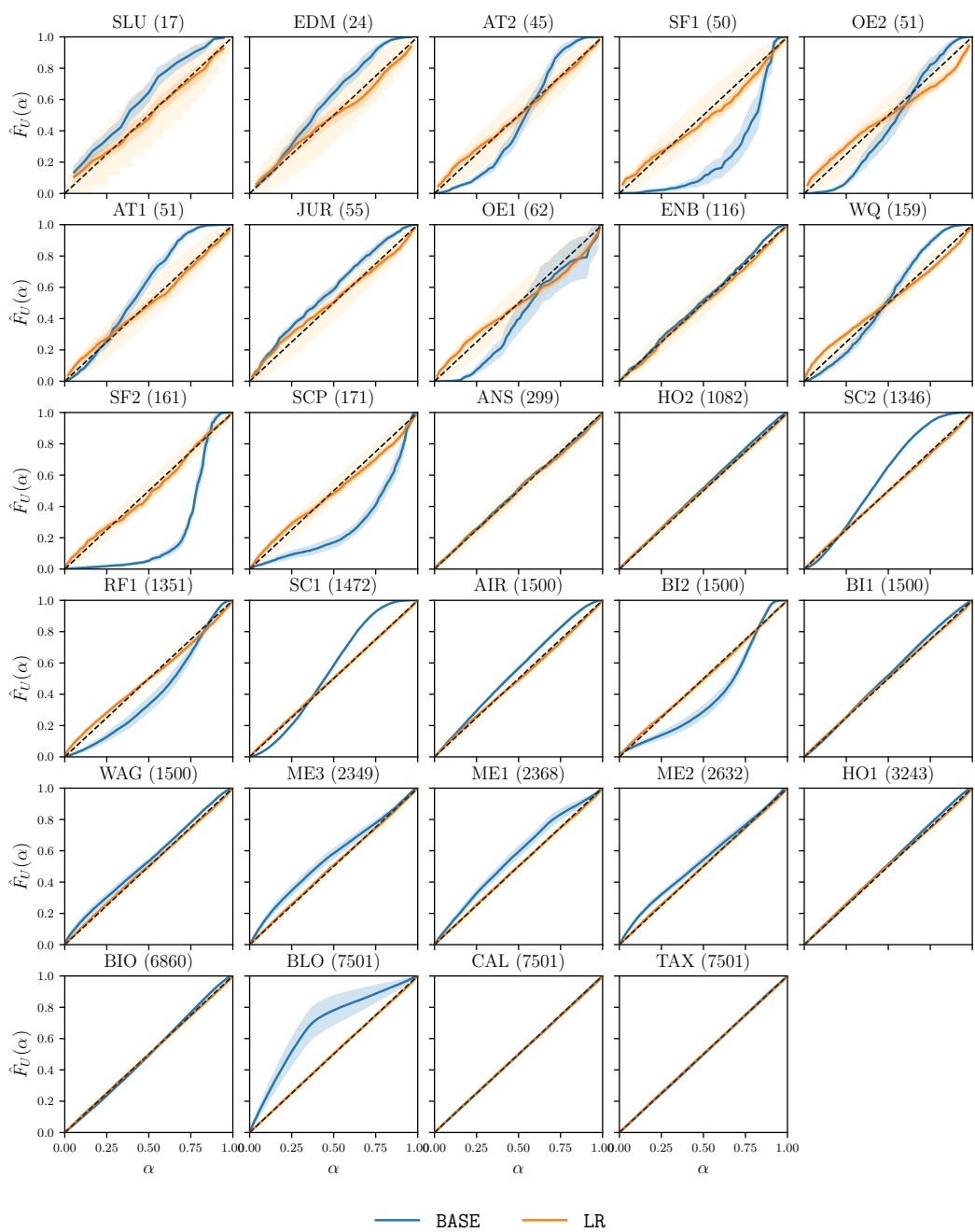

Figure 8: Latent calibration diagrams

Figure 9 shows reliability diagrams for HDR calibration. These diagrams plot nominal probability levels $\alpha$ against empirical probabilities $\hat{F}_{\hat{U}}(\alpha)$, where $\hat{U} = \text{HPD}_{\hat{f}_{Y|X}}(Y)$ are the HDR pre-rank values from the test set. Again, the BASE model frequently shows miscalibration. Both LR and HDR-R improve HDR calibration, though for LR this improvement is a beneficial side effect rather than a direct optimization target, unlike its consistent improvement of latent calibration shown in Figure 8.

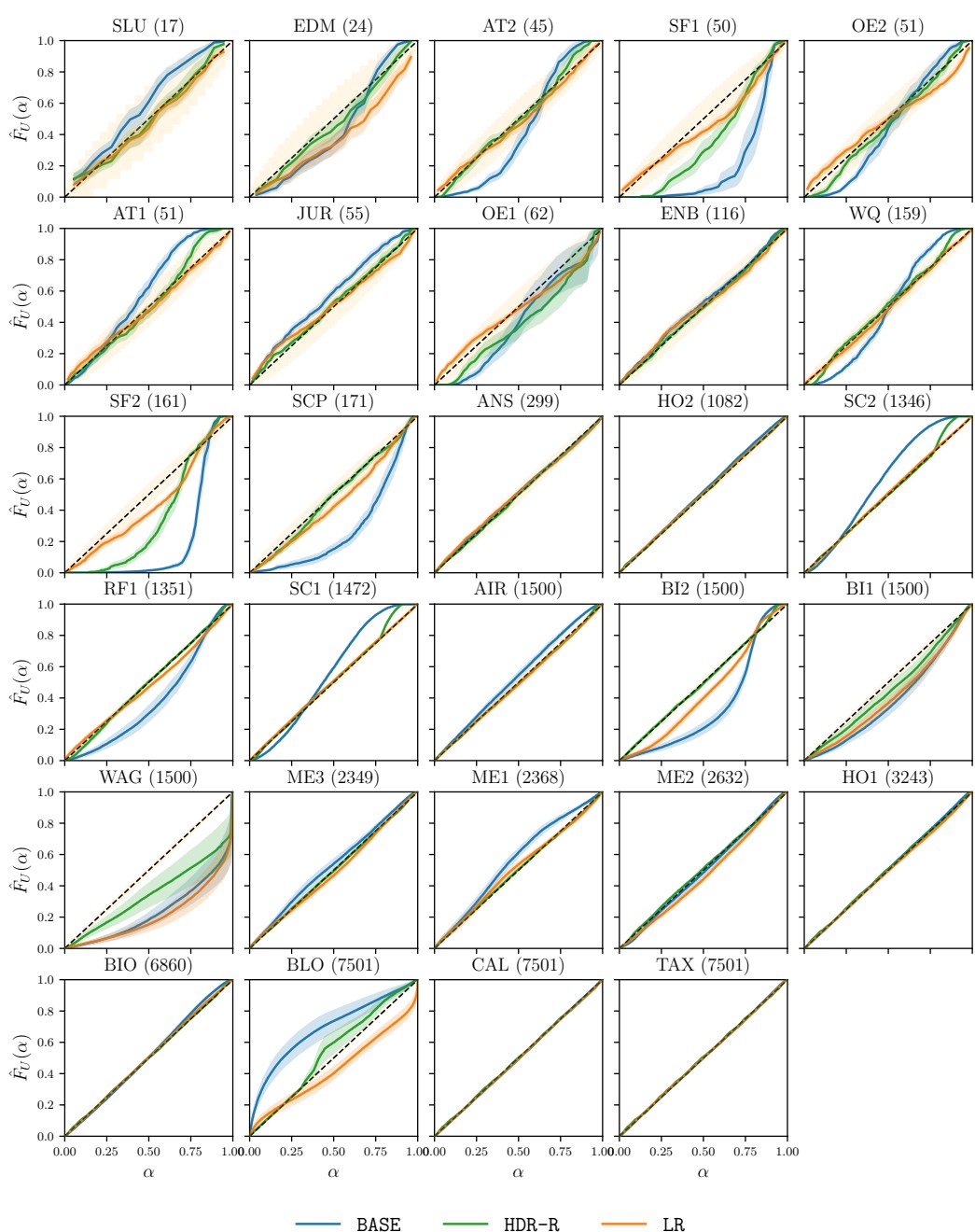

Figure 9: HDR-calibration diagrams

# I  Base predictors

This section provides details on the base predictors considered in this paper.

## I.1  Convex potential flows

Convex potential flows (Huang et al., 2020) parameterize the bijective transformation $\hat{T}$ via a strongly convex potential whose gradient yields the inverse map. Given $x \in \mathcal{X}$, let the model define a scalar

potential $\hat{V} : \mathcal{Y} \times \mathcal{X} \to \mathbb{R}$ where $\hat{V}(\cdot; x)$ is strongly convex for every fixed $x$. The associated transformation is the gradient

$$\hat{T}^{-1}(y; x) = \nabla_y \hat{V}(y; x) \in \mathbb{R}^d.$$

To ensure convexity in $y$, $\hat{V}(\cdot; x)$ is parameterized by an *input-convex neural network* (ICNN, Amos et al., 2017). A small quadratic term $\frac{\alpha}{2}\|y\|^2$ enforces strong convexity of $\hat{V}(\cdot; x)$.

Since $\hat{V}(\cdot; x)$ is strongly convex, $\hat{T}^{-1}(\cdot; x)$ is a bijection $\mathcal{Y} \to \mathcal{Z}$ with inverse $\hat{T}(\cdot; x)$. Generating new conditional samples requires inverting $\hat{T}^{-1}(\cdot; x)$. Given $z \in \mathcal{Z}$, one recovers $\hat{T}(z; x)$ as the unique minimizer of the convex objective

$$\hat{T}(z; x) \in \operatorname*{arg\,min}_{y \in \mathcal{Y}} \left\{ \hat{V}(y; x) - z^\top y \right\}. \tag{29}$$

Indeed, since $\hat{V}(\cdot; x)$ is differentiable and strongly convex, the minimum is attained when

$$\nabla_y \left( \hat{V}(y; x) - z^\top y \right) = 0 \iff \nabla_y \hat{V}(y; x) = z \iff \hat{T}^{-1}(y; x) = z \iff y = \hat{T}(z; x). \tag{30}$$

In practice, (29) can be solved efficiently with gradient-based convex optimization (e.g., L-BFGS), and strong convexity ensures convergence to a unique solution.

The Hessian $\nabla_y^2 \hat{V}(y; x)$ of is positive definite, with

$$\left| \det \nabla_y \hat{T}^{-1}(y; x) \right| = \det \left( \nabla_y^2 \hat{V}(y; x) \right).$$

One approach to compute this determinant is by explicitly forming the Hessian $H = \nabla_y^2 \hat{V}(y; x)$. Concretely, $\hat{T}^{-1}(y; x) = \nabla_y \hat{V}(y; x)$ is first evaluated with a single forward pass and backpropagation, and then the Hessian is formed as

$$H = \left( \frac{\partial \hat{T}^{-1}(y; x)}{\partial y_1}, \ldots, \frac{\partial \hat{T}^{-1}(y; x)}{\partial y_d} \right) \in \mathbb{R}^{d \times d},$$

requiring $d$ additional backpropagations. Finally, the determinant can be computed explicitly in $O(d^3)$ (e.g., via Cholesky). Hence, the overall computational complexity is $O\left( d\, C_{\text{backprop}} + d^3 \right)$ where $C_{\text{backprop}}$ denotes the cost of a single backpropagation. This brute-force approach is practical for small $d$ but becomes prohibitive as $d$ grows; more efficient strategies for large $d$ are discussed in Huang et al., 2020.

## I.2 Masked autoregressive flows (MAFs)

Masked autoregressive flows (Papamakarios, Pavlakou, et al., 2017) implement $\hat{T}^{-1}$ as an autoregressive affine transformation. For each coordinate $i \in [d]$, the model uses two autoregressive NNs (e.g. MADE, (Germain et al., 2015)) to parameterize

$$\hat{\mu}_i : \mathbb{R}^{i-1} \times \mathcal{X} \to \mathbb{R}, \qquad \hat{\rho}_i : \mathbb{R}^{i-1} \times \mathcal{X} \to \mathbb{R},$$

and $\hat{\sigma}_i(y_{<i}, x) = \exp(\hat{\rho}_i(y_{<i}, x))$ ensures positive outputs. The masking mechanism ensures that $\hat{\mu}_i(y_{<i}, x)$ and $\hat{\sigma}_i(y_{<i}, x)$ depend only on the preceding coordinates $y_{<i} = (y_1, \ldots, y_{i-1})$ and the conditioning variable $x$.

The inverse transformation for each coordinate is then

$$z_i = \frac{y_i - \hat{\mu}_i(y_{<i}; x)}{\hat{\sigma}_i(y_{<i}; x)}, \qquad i = 1, \ldots, d.$$

Since this mapping is triangular in $y$, the Jacobian of $\hat{T}^{-1}$ is also triangular, which makes the determinant computation efficient:

$$\left| \det \nabla_y \hat{T}^{-1}(y; x) \right| = \prod_{i=1}^d \frac{1}{\hat{\sigma}_i(y_{<i}; x)}.$$

### I.3 Flow matching (FM)

Flow matching (Lipman et al., 2022) is a recent generative modeling paradigm which has rapidly been gaining popularity. The key motivation behind flow matching is to combine the strengths of normalizing flows (NFs) and diffusion models while alleviating their main limitations. NFs enable exact likelihood estimation and efficient sampling but often suffer from limited expressiveness due to architectural constraints. Diffusion models, on the other hand, offer remarkable expressiveness and stability but typically require slow iterative sampling and do not provide tractable likelihoods. Flow matching addresses these issues by framing generative modeling as the learning of a continuous-time flow that transports noise to data, enabling both efficient training and fast sampling while retaining theoretical connections to likelihood-based methods.

Given $x \in \mathcal{X}$, we model the conditional predictive PDF $\hat{f}$ using a transformation defined by the ordinary differential equation (ODE) $\frac{d\tilde{y}}{dt} = \hat{v}(t, \tilde{y}, x)$, with a NN-parameterized vector field $\hat{v} : [0, 1] \times \mathcal{Y} \times \mathcal{X} \to \mathcal{Y}$. Training uses the straight-line interpolant between latent $z \sim \mathcal{N}(0, I)$ and data $y \sim \hat{f}(\cdot \mid x)$,

$$\tilde{y}(t) = (1 - t) z + ty,$$

whose target velocity is constant with respect to $t$:

$$\frac{d}{dt}\tilde{y}(t) = y - z.$$

The Conditional Flow Matching objective is

$$\min_{\hat{v}} \ \mathbb{E} \left\| \hat{v}(t, (1 - t) Z + tY, X) - (Y - Z) \right\|^2 \tag{31}$$

where the expectation is over $t \sim \mathcal{U}(0, 1)$, $(X, Y) \sim P_{X,Y}$, and $Z \sim \mathcal{N}(0, I)$. After training, forward numerical integration generates samples from $\hat{f}_{Y|X=x}$ (Chen et al., 2018):

$$\hat{T}(z; x) = \tilde{y}(1) = z + \int_0^1 \hat{v}(t, \tilde{y}(t), x) \, dt, \qquad \tilde{y}(0) = z. \tag{32}$$

Reverse-time integration encodes $y$ into its latent $z$:

$$\hat{T}^{-1}(y; x) = \tilde{y}(0) = y + \int_1^0 \hat{v}(t, \tilde{y}(t), x) \, dt, \qquad \tilde{y}(1) = y. \tag{33}$$

For $y \in \mathcal{Y}$, set $z = \hat{T}^{-1}(y; x)$. The log-likelihood follows from the instantaneous change-of-variables formula along the unique ODE path $\tilde{y}(t)$ with $\tilde{y}(0) = z$ and $\tilde{y}(1) = y$:

$$\log \hat{f}_{Y|X=x}(y) = \log f_Z(z) - \int_0^1 \text{Tr}(\nabla_{\tilde{y}} \hat{v}(t, \tilde{y}(t), x)) \, dt. \tag{34}$$

The trace of the Jacobian can be computed using $d$ backpropagations, which can be prohibitive if $d$ is large. It can also be efficiently approximated using Hutchinson's estimator:

$$\text{Tr}(\nabla_{\tilde{y}} \hat{v}(t, \tilde{y}(t), x)) = \mathbb{E}_{\epsilon \sim \mathcal{N}(0, I)} \left[ \epsilon^\top (\nabla_{\tilde{y}} \hat{v}(t, \tilde{y}(t), x)) \epsilon \right] \approx \frac{1}{K} \sum_{k=1}^K \epsilon_k^\top (\nabla_{\tilde{y}} \hat{v}(t, \tilde{y}(t), x)) \epsilon_k,$$
$$\tag{35}$$

with independent $\epsilon_k \sim \mathcal{N}(0, I)$, which enables practical likelihood computation.

## J  Additional results with convex potential flows

### J.1  Investigating the NLL performance gain

To investigate the source of the NLL improvement, we consider the decomposition of the NLL of the recalibrated model:

$$-\log \hat{f}'_{Y|X=x}(y) = -\log f_Z(z) - \log \left| \det (\nabla_z R(z)) \right|^{-1} - \log \left| \det \left( \nabla_y \hat{T}^{-1}(y; x) \right) \right| \tag{36}$$

with $z' = \hat{T}^{-1}(y; x)$ and $z = R^{-1}(z')$. The third term is identical for both BASE and LR. All reported terms are averaged over the test set and over 10 runs.

Table 5: Analysis of the NLL of LR compared to BASE with the terms described in (36).

| | BASE $-\log f_Z(z')$ | LR $-\log f_Z(z)$ | LR $-\log|\det(\nabla_z R(z))|^{-1}$ | BASE $-\log \hat{f}_{Y|X=x}(y)$ | LR $-\log \hat{f}'_{Y|X=x}(y)$ |
|---|---|---|---|---|---|
| SLU | 5.49 | 4.35 | 0.828 | 2.61 | 2.29 |
| EDM | 4.94 | 2.81 | 2.04 | -0.0350 | -0.123 |
| AT2 | 11.7 | 8.65 | 0.889 | 4.05 | 1.86 |
| SF1 | 11.7 | 4.33 | 2.53 | 4.41 | -0.381 |
| OE2 | 54.6 | 22.6 | 0.896 | 37.6 | 6.46 |
| AT1 | 11.6 | 8.47 | 1.56 | 1.63 | 0.0783 |
| JUR | 5.17 | 4.24 | 0.638 | 2.16 | 1.87 |
| OE1 | 98.0 | 22.6 | -1.65 | 80.0 | 2.93 |
| ENB | 3.08 | 2.80 | 0.244 | -1.08 | -1.12 |
| WQ | 74.3 | 19.9 | 1.51 | 60.8 | 7.95 |
| SF2 | 10.9 | 4.29 | -1.10 | -3.37 | -11.1 |
| SCP | 32.1 | 4.19 | -0.757 | 20.1 | -8.55 |
| ANS | 2.82 | 2.78 | 0.0700 | 1.76 | 1.79 |
| HO2 | 5.81 | 5.67 | 0.143 | 2.39 | 2.38 |
| SC2 | 25.1 | 22.6 | 1.80 | 0.795 | 0.189 |
| RF1 | 9.70e+02 | 11.3 | -0.0264 | 9.54e+02 | -4.50 |
| SC1 | 24.9 | 22.7 | 1.39 | -1.86 | -2.63 |
| AIR | 9.15 | 8.47 | 0.565 | 3.03 | 2.91 |
| BI2 | 6.99 | 5.68 | -0.208 | -11.5 | -13.0 |
| BI1 | 3.10 | 2.82 | 0.151 | -2.27 | -2.40 |
| WAG | 3.09 | 2.82 | 0.184 | -3.25 | -3.34 |
| ME3 | 3.07 | 2.83 | 0.146 | -2.60 | -2.69 |
| ME1 | 3.21 | 2.81 | 0.225 | -2.00 | -2.17 |
| ME2 | 3.19 | 2.82 | 0.143 | -3.00 | -3.22 |
| HO1 | 3.03 | 2.82 | 0.144 | -0.299 | -0.364 |
| BIO | 3.08 | 2.82 | 0.128 | -1.12 | -1.25 |
| BLO | 9.18 | 2.82 | 1.14 | 3.11 | -2.11 |
| CAL | 2.84 | 2.82 | 0.0255 | 0.575 | 0.575 |
| TAX | 2.85 | 2.83 | 0.0208 | 1.53 | 1.53 |

The table reveals a clear pattern. The NLL improvement from LR is primarily driven by the first term. By radially transforming the latent codes $z'$ to new points $z$ that are more consistent with the base density $f_Z$, the latent density term $-\log f_Z(z)$ is significantly reduced. The recalibration Jacobian (second term) typically adds a small penalty (increases NLL), but this is almost always outweighed by the large gains from the first term. This confirms that LR works by finding more "plausible" latent codes for the observed data under the base latent distribution.

## J.2 Computational efficiency

The difference in computation time can be measured in two aspects:

- For calibration, the computational complexity of HDR-R is $O(MFn)$ and LR is $O(Rn)$ where $M = 100$ corresponds to the number of samples of HDR-R per instance, $F$ the time for the forward mapping $\hat{T}$ and $R$ the time for the reverse mapping $\hat{T}^{-1}$.

- For inference, it is a bit more subtle. Given a test insance $x$, HDR-R requires to sample at least $M$ times ($O(MF)$) to obtain a recalibrated sample, which can be a weakness, e.g., if only one conditional sample is needed. LR only incurs a low fixed cost $C$ for evaluating the recalibration map ($O(C + F)$). Thus, the inference time is not directly comparable.

We report the calibration time of HDR-R and LR in seconds on the largest datasets using the convex potential flow model and averaged over 10 runs.

Table 6: Calibration times (part 1)

| Method | HO2 | SC2 | RF1 | SC1 | AIR | BI2 | BI1 | WAG |
|---|---|---|---|---|---|---|---|---|
| HDR-R | 1.56 | 2.78 | 4.90 | 2.57 | 4.86 | 18.80 | 8.80 | 15.00 |
| LR | 0.232 | 0.185 | 0.156 | 0.187 | 0.149 | 0.142 | 0.133 | 0.133 |

Table 7: Calibration times (part 2)

| Method | ME3 | ME1 | ME2 | HO1 | BIO | BLO | CAL | TAX |
|---|---|---|---|---|---|---|---|---|
| HDR-R | 9.98 | 12.70 | 19.30 | 7.19 | 37.70 | 168.00 | 8.53 | 10.50 |
| LR | 0.152 | 0.153 | 0.163 | 0.192 | 0.290 | 0.328 | 0.482 | 0.324 |

On CIFAR-10 with TarFlow, the time difference is larger and can be prohibitive for HDR-R:

| Method | CIFAR-10 |
|---|---|
| HDR-R | 183182 |
| LR | 1259 |

## J.3 Discriminative ability of the energy score and NLL

For a comprehensive evaluation of LR, we report the ES in addition to the NLL. While LR often leads to improved NLL, the ES remains largely unchanged. We hypothesize that this stems from the score's fundamental limitations in discriminative ability.

**Theoretical considerations.** As established in Pinson and Tastu, 2013 and corroborated by Alexander et al., 2022, the ES is sensitive to shifts in the mean but notoriously insensitive to misspecifications in variance, correlation, and overall dependency structure. LR is a post-hoc procedure that primarily corrects the shape and spread of the predictive distribution. Therefore, the ES is fundamentally ill-suited to capture the specific improvements LR provides.

In contrast, the NLL is uniquely suited for this evaluation. As the only local strictly proper scoring rule, its value depends only on the probability density at the precise location of the observed outcome (Du, 2021). This locality makes it highly discerning of the very improvements LR makes to the distributional shape, which is why we observe significant and consistent NLL reductions.

**Empirical illustration.** To provide a clear, empirical illustration, we designed a controlled synthetic experiment based on the dataset in Figure 1. The goal here is to isolate this specific property of the scoring rules in a setting free from the confounding variables of complex, real-world data.

We use an oracle predictor that knows the true data-generating distribution from Figure 1 for everything except the spread around the arc, which is controlled by a standard deviation parameter $\sigma$. In Table 8, we then evaluate the predictor's NLL and ES (based on 100 samples) as we vary its estimate of $\sigma$. The metrics are averaged over 10 runs, and the true value is $\sigma = 0.05$.

This experiment clearly illustrates the issue:

- The NLL shows a sharp, clear minimum at the true value of $\sigma = 0.05$, correctly identifying the best model.
- The ES remains almost completely flat for a wide range of $\sigma$ values (from 0.01 to 0.10). It fails to reliably distinguish a model with the correct variance from one that is substantially over- or under-confident.

This insensitivity is so profound that detecting a statistically significant signal with the ES requires an impractically large number of samples. The table below shows that only with 5000 runs does the ES minimum align with the true $\sigma$, and even then the differences are minuscule:

This controlled experiment, therefore, proposes an explanation for the insentivity of the ES to LR.

Table 8: Metrics averaged over 10 runs, with standard error

| $\sigma$ | NLL | ES |
|---|---|---|
| 0.01 | $12.01_{0.195}$ | $0.8733_{0.00298}$ |
| 0.03 | $1.274_{0.0222}$ | $0.8724_{0.00267}$ |
| 0.04 | $0.9232_{0.0129}$ | $\mathbf{0.8723_{0.00295}}$ |
| 0.05 (True) | $\mathbf{0.8557_{0.00870}}$ | $0.8740_{0.00171}$ |
| 0.06 | $0.8781_{0.00629}$ | $0.8741_{0.00182}$ |
| 0.07 | $0.9365_{0.00483}$ | $0.8753_{0.00214}$ |
| 0.10 | $1.161_{0.00272}$ | $0.8741_{0.00163}$ |
| 0.20 | $1.774_{0.00106}$ | $0.8782_{0.00176}$ |

Table 9: Metrics averaged over 5000 runs, with standard error

| $\sigma$ | NLL | ES |
|---|---|---|
| 0.01 | $11.91_{0.00773}$ | $0.8754_{0.000139}$ |
| 0.03 | $1.263_{0.000868}$ | $0.8751_{0.000139}$ |
| 0.04 | $0.9177_{0.000495}$ | $0.8751_{0.000139}$ |
| 0.05 (True) | $\mathbf{0.8487_{0.000323}}$ | $\mathbf{0.8750_{0.000139}}$ |
| 0.06 | $0.8732_{0.000231}$ | $0.8751_{0.000138}$ |
| 0.07 | $0.9329_{0.000176}$ | $0.8751_{0.000138}$ |
| 0.10 | $1.159_{0.000104}$ | $0.8756_{0.000137}$ |
| 0.20 | $1.774_{6.91e-05}$ | $0.8797_{0.000133}$ |

## J.4 Additional tables

For reference, Tables 10 and 11 provide the precise mean values and standard errors for NLL, Energy Score, L-ECE, and HDR-ECE across all tabular datasets when using convex potential flows as the base predictor. For each metric and dataset, all values that are statistically indistinguishable to the best value according to a Z-test at significance level 0.1 are highlighted in bold.

Table 10: Full comparative table, using a convex potential flow model.

| | NLL | | Energy score | | |
| | BASE | LR | BASE | HDR–R | LR |
|---|---|---|---|---|---|
| SLU | $\mathbf{2.61_{0.19}}$ | $\mathbf{2.29_{0.14}}$ | $\mathbf{0.791_{0.038}}$ | $\mathbf{0.795_{0.033}}$ | $\mathbf{0.785_{0.033}}$ |
| EDM | $\mathbf{-0.0350_{0.46}}$ | $\mathbf{-0.123_{1.3}}$ | $0.647_{0.049}$ | $\mathbf{0.648_{0.050}}$ | $\mathbf{0.635_{0.044}}$ |
| AT2 | $4.05_{0.88}$ | $\mathbf{1.86_{0.42}}$ | $0.861_{0.044}$ | $0.870_{0.044}$ | $\mathbf{0.862_{0.042}}$ |
| SF1 | $4.41_{3.2}$ | $\mathbf{-0.381_{3.6}}$ | $0.673_{0.086}$ | $\mathbf{0.639_{0.093}}$ | $0.670_{0.085}$ |
| OE2 | $37.6_{3.0e+01}$ | $\mathbf{6.46_{1.3}}$ | $\mathbf{1.25_{0.083}}$ | $\mathbf{1.26_{0.083}}$ | $\mathbf{1.26_{0.085}}$ |
| AT1 | $1.63_{0.46}$ | $\mathbf{0.0783_{0.29}}$ | $\mathbf{0.582_{0.032}}$ | $\mathbf{0.587_{0.031}}$ | $\mathbf{0.591_{0.030}}$ |
| JUR | $2.16_{0.24}$ | $\mathbf{1.87_{0.15}}$ | $\mathbf{0.617_{0.034}}$ | $\mathbf{0.618_{0.033}}$ | $\mathbf{0.618_{0.034}}$ |
| OE1 | $80.0_{6.9e+01}$ | $\mathbf{2.93_{0.71}}$ | $\mathbf{1.23_{0.15}}$ | $\mathbf{1.23_{0.15}}$ | $\mathbf{1.17_{0.15}}$ |
| ENB | $\mathbf{-1.08_{0.11}}$ | $\mathbf{-1.12_{0.10}}$ | $\mathbf{0.249_{0.010}}$ | $\mathbf{0.250_{0.010}}$ | $\mathbf{0.249_{0.010}}$ |
| WQ | $60.8_{3.7e+01}$ | $\mathbf{7.95_{3.6}}$ | $\mathbf{2.47_{0.025}}$ | $2.49_{0.024}$ | $\mathbf{2.47_{0.024}}$ |
| SF2 | $-3.37_{3.0}$ | $\mathbf{-11.1_{0.63}}$ | $\mathbf{0.587_{0.044}}$ | $0.593_{0.046}$ | $\mathbf{0.598_{0.045}}$ |
| SCP | $20.1_{2.6e+01}$ | $\mathbf{-8.55_{0.49}}$ | $0.389_{0.094}$ | $\mathbf{0.383_{0.095}}$ | $\mathbf{0.382_{0.095}}$ |
| ANS | $\mathbf{1.76_{0.022}}$ | $1.79_{0.020}$ | $\mathbf{0.529_{0.0052}}$ | $0.531_{0.0047}$ | $\mathbf{0.529_{0.0053}}$ |
| HO2 | $2.39_{0.034}$ | $\mathbf{2.38_{0.035}}$ | $\mathbf{0.862_{0.0076}}$ | $0.866_{0.0075}$ | $\mathbf{0.862_{0.0076}}$ |
| SC2 | $0.795_{0.17}$ | $\mathbf{0.189_{0.15}}$ | $\mathbf{1.25_{0.011}}$ | $1.28_{0.012}$ | $1.26_{0.011}$ |
| RF1 | $9.54e+02_{6.8e+02}$ | $\mathbf{-4.50_{1.5}}$ | $0.534_{0.073}$ | $\mathbf{0.528_{0.071}}$ | $\mathbf{0.529_{0.072}}$ |
| SC1 | $-1.86_{0.080}$ | $\mathbf{-2.63_{0.075}}$ | $0.824_{0.0047}$ | $\mathbf{0.833_{0.0045}}$ | $0.825_{0.0045}$ |
| AIR | $\mathbf{3.03_{0.30}}$ | $\mathbf{2.91_{0.30}}$ | $\mathbf{1.17_{0.0086}}$ | $1.19_{0.0090}$ | $1.18_{0.0084}$ |
| BI2 | $\mathbf{-11.5_{0.71}}$ | $\mathbf{-13.0_{0.61}}$ | $0.833_{0.013}$ | $0.848_{0.015}$ | $\mathbf{0.834_{0.014}}$ |
| BI1 | $\mathbf{-2.27_{0.26}}$ | $\mathbf{-2.40_{0.26}}$ | $\mathbf{0.708_{0.0056}}$ | $0.711_{0.0053}$ | $\mathbf{0.708_{0.0057}}$ |
| WAG | $\mathbf{-3.25_{0.31}}$ | $\mathbf{-3.34_{0.32}}$ | $\mathbf{0.802_{0.048}}$ | $\mathbf{0.803_{0.043}}$ | $\mathbf{0.805_{0.046}}$ |
| ME3 | $\mathbf{-2.60_{0.13}}$ | $\mathbf{-2.69_{0.12}}$ | $\mathbf{0.358_{0.0082}}$ | $0.362_{0.0082}$ | $\mathbf{0.360_{0.0083}}$ |
| ME1 | $\mathbf{-2.00_{0.13}}$ | $\mathbf{-2.17_{0.13}}$ | $\mathbf{0.357_{0.0075}}$ | $0.370_{0.0095}$ | $\mathbf{0.365_{0.0087}}$ |
| ME2 | $\mathbf{-3.00_{0.12}}$ | $\mathbf{-3.22_{0.13}}$ | $\mathbf{0.361_{0.0041}}$ | $0.362_{0.0039}$ | $\mathbf{0.362_{0.0040}}$ |
| HO1 | $\mathbf{-0.299_{0.038}}$ | $\mathbf{-0.364_{0.029}}$ | $\mathbf{0.346_{0.0074}}$ | $0.351_{0.0079}$ | $\mathbf{0.340_{0.012}}$ |
| BIO | $-1.12_{0.072}$ | $\mathbf{-1.25_{0.019}}$ | $0.207_{0.0038}$ | $0.210_{0.0039}$ | $\mathbf{0.204_{0.0051}}$ |
| BLO | $3.11_{2.8}$ | $\mathbf{-2.11_{0.25}}$ | $\mathbf{0.305_{0.029}}$ | $0.365_{0.037}$ | $0.473_{0.081}$ |
| CAL | $\mathbf{0.575_{0.0097}}$ | $\mathbf{0.575_{0.0088}}$ | $\mathbf{0.419_{0.0014}}$ | $0.421_{0.0015}$ | $\mathbf{0.419_{0.0014}}$ |
| TAX | $\mathbf{1.53_{0.0069}}$ | $\mathbf{1.53_{0.0068}}$ | $\mathbf{0.692_{0.0019}}$ | $0.696_{0.0021}$ | $0.690_{0.0027}$ |

Table 11: Full comparative table, using a convex potential flow model.

| | L-ECE | | HDR-ECE | | |
| | BASE | LR | BASE | HDR-R | LR |
|---|---|---|---|---|---|
| SLU | **$0.146_{0.026}$** | **$0.106_{0.016}$** | **$0.129_{0.022}$** | **$0.116_{0.014}$** | **$0.102_{0.013}$** |
| EDM | **$0.122_{0.016}$** | **$0.0905_{0.016}$** | **$0.128_{0.014}$** | **$0.101_{0.019}$** | $0.169_{0.014}$ |
| AT2 | $0.129_{0.0076}$ | **$0.0637_{0.010}$** | $0.149_{0.0094}$ | **$0.0817_{0.0095}$** | **$0.0688_{0.012}$** |
| SF1 | $0.279_{0.026}$ | **$0.0701_{0.0082}$** | $0.321_{0.022}$ | $0.171_{0.019}$ | **$0.0786_{0.0096}$** |
| OE2 | $0.136_{0.011}$ | **$0.0785_{0.0076}$** | $0.129_{0.0098}$ | **$0.0768_{0.012}$** | **$0.0766_{0.0079}$** |
| AT1 | $0.124_{0.013}$ | **$0.0668_{0.0098}$** | $0.125_{0.012}$ | $0.0906_{0.0095}$ | **$0.0643_{0.012}$** |
| JUR | $0.0883_{0.011}$ | **$0.0520_{0.0052}$** | $0.0866_{0.011}$ | **$0.0515_{0.0063}$** | **$0.0537_{0.0049}$** |
| OE1 | $0.197_{0.044}$ | **$0.0589_{0.0068}$** | $0.195_{0.044}$ | **$0.144_{0.049}$** | **$0.0636_{0.0074}$** |
| ENB | **$0.0272_{0.0034}$** | **$0.0292_{0.0034}$** | **$0.0375_{0.0049}$** | **$0.0347_{0.0039}$** | **$0.0397_{0.0061}$** |
| WQ | $0.0883_{0.0048}$ | **$0.0479_{0.0044}$** | $0.0975_{0.0032}$ | **$0.0443_{0.0060}$** | **$0.0432_{0.0086}$** |
| SF2 | $0.272_{0.0084}$ | **$0.0434_{0.0050}$** | $0.312_{0.0061}$ | $0.164_{0.015}$ | **$0.0797_{0.010}$** |
| SCP | $0.218_{0.025}$ | **$0.0502_{0.0060}$** | $0.223_{0.025}$ | **$0.0543_{0.010}$** | **$0.0614_{0.0093}$** |
| ANS | **$0.0190_{0.0029}$** | **$0.0254_{0.0036}$** | **$0.0199_{0.0030}$** | **$0.0234_{0.0026}$** | **$0.0249_{0.0037}$** |
| HO2 | **$0.0162_{0.0024}$** | **$0.0158_{0.0018}$** | $0.0179_{0.0023}$ | **$0.0122_{0.00080}$** | **$0.0121_{0.0017}$** |
| SC2 | $0.101_{0.0034}$ | **$0.0122_{0.00087}$** | $0.105_{0.0030}$ | $0.0223_{0.00081}$ | **$0.0117_{0.0011}$** |
| RF1 | $0.112_{0.021}$ | **$0.0260_{0.0049}$** | $0.129_{0.024}$ | **$0.0193_{0.0034}$** | $0.0288_{0.0045}$ |
| SC1 | $0.0845_{0.0015}$ | **$0.0106_{0.0013}$** | $0.0857_{0.0017}$ | $0.0229_{0.0011}$ | **$0.0116_{0.0016}$** |
| AIR | $0.0527_{0.0059}$ | **$0.0162_{0.0010}$** | $0.0449_{0.0083}$ | **$0.0160_{0.0012}$** | $0.0286_{0.0052}$ |
| BI2 | $0.122_{0.016}$ | **$0.0170_{0.0022}$** | $0.167_{0.019}$ | **$0.0219_{0.0023}$** | $0.0731_{0.011}$ |
| BI1 | $0.0279_{0.0023}$ | **$0.0102_{0.00084}$** | **$0.111_{0.026}$** | **$0.0580_{0.024}$** | **$0.0923_{0.023}$** |
| WAG | $0.0478_{0.0082}$ | **$0.0137_{0.0018}$** | **$0.267_{0.050}$** | **$0.161_{0.071}$** | **$0.290_{0.046}$** |
| ME3 | $0.0732_{0.012}$ | **$0.00957_{0.00084}$** | $0.0520_{0.010}$ | **$0.00899_{0.00064}$** | $0.0323_{0.0042}$ |
| ME1 | $0.0853_{0.010}$ | **$0.0103_{0.00078}$** | $0.0779_{0.010}$ | **$0.00976_{0.00092}$** | $0.0432_{0.0050}$ |
| ME2 | $0.0497_{0.0099}$ | **$0.0101_{0.00066}$** | $0.0415_{0.0061}$ | **$0.0136_{0.0023}$** | $0.0417_{0.0061}$ |
| HO1 | $0.0166_{0.0023}$ | **$0.00950_{0.0013}$** | **$0.0103_{0.0014}$** | **$0.00782_{0.00071}$** | **$0.0109_{0.0018}$** |
| BIO | $0.0178_{0.0019}$ | **$0.00561_{0.00076}$** | $0.0220_{0.0027}$ | **$0.00667_{0.00051}$** | **$0.00727_{0.0015}$** |
| BLO | $0.231_{0.036}$ | **$0.00735_{0.0010}$** | $0.207_{0.040}$ | **$0.0485_{0.024}$** | $0.114_{0.014}$ |
| CAL | $0.00749_{0.00096}$ | **$0.00502_{0.00074}$** | $0.00760_{0.00079}$ | **$0.00555_{0.00026}$** | **$0.00543_{0.00064}$** |
| TAX | $0.00949_{0.0013}$ | **$0.00522_{0.00053}$** | **$0.00584_{0.00063}$** | **$0.00684_{0.00050}$** | **$0.00561_{0.00086}$** |

# K  Results with MAFs

For completeness, this section reports results using a MAF (Papamakarios, Pavlakou, et al., 2017) as the base predictor. The architecture consists of stacked flow layers, where each layer's conditioner is a masked autoencoder (Germain et al., 2015) parameterizing rational quadratic spline transformations (Durkan et al., 2019). We tune hyperparameters using grid search. The number of stacked flows is chosen from $[3, 5, 8]$, the number of hidden units per flow from $[32, 64]$, and the number of hidden layers per flow from $[2, 3]$. The learning rate is selected from $[5 \times 10^{-3}, 10^{-3}]$. Each flow learns a rational quadratic spline transformation.

The findings, illustrated in Figure 10 and Figure 11 (and detailed in Tables 12 and 13), are consistent with the main tabular results reported in Section 5. Specifically, LR provides notable improvements in L-ECE, NLL, and HDR-ECE, while achieving an energy score comparable to that of the BASE model.

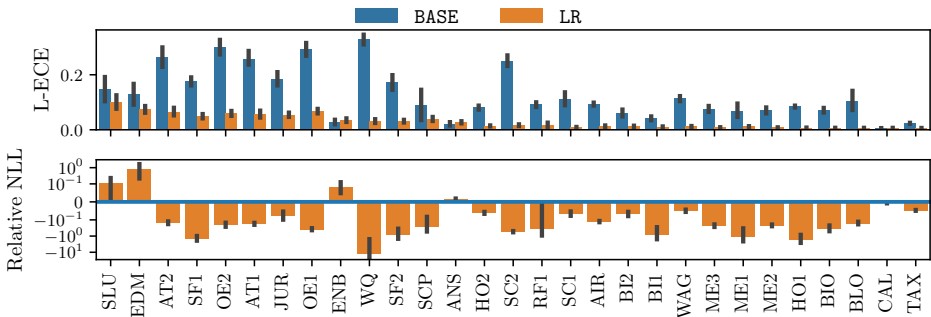

Figure 10: Latent calibration and NLL on datasets sorted by size, using a MAF model.

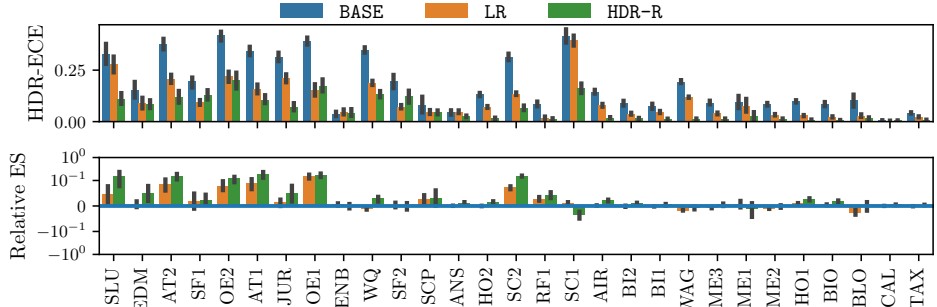

Figure 11: Latent calibration and NLL on datasets sorted by size, using a MAF model.

Table 12: Full comparative table, using a MAF model.

| | NLL | | Energy score | | |
| | BASE | LR | BASE | HDR-R | LR |
|---|---|---|---|---|---|
| SLU | **4.50$_{0.37}$** | 5.08$_{0.52}$ | **0.831$_{0.051}$** | 0.945$_{0.034}$ | 0.858$_{0.025}$ |
| EDM | **0.202$_{0.30}$** | 0.590$_{0.35}$ | 0.577$_{0.035}$ | 0.602$_{0.033}$ | 0.578$_{0.022}$ |
| AT2 | 7.45$_{0.52}$ | **6.22$_{0.24}$** | **0.970$_{0.054}$** | 1.11$_{0.044}$ | 1.05$_{0.029}$ |
| SF1 | -1.47$_{0.39}$ | **-3.42$_{0.26}$** | **0.698$_{0.087}$** | 0.710$_{0.085}$ | 0.700$_{0.053}$ |
| OE2 | 22.4$_{2.6}$ | **16.8$_{0.90}$** | **1.67$_{0.13}$** | 1.85$_{0.13}$ | 1.79$_{0.087}$ |
| AT1 | **4.63$_{0.61}$** | 3.80$_{0.32}$ | **0.698$_{0.049}$** | 0.815$_{0.043}$ | 0.754$_{0.029}$ |
| JUR | **3.99$_{0.29}$** | 3.65$_{0.15}$ | **0.680$_{0.037}$** | 0.710$_{0.031}$ | 0.686$_{0.022}$ |
| OE1 | 16.9$_{2.7}$ | **9.68$_{0.75}$** | **1.31$_{0.13}$** | 1.51$_{0.12}$ | 1.48$_{0.080}$ |
| ENB | **-0.939$_{0.10}$** | -0.877$_{0.071}$ | 0.273$_{0.0080}$ | **0.273$_{0.0077}$** | 0.275$_{0.0054}$ |
| WQ | 0.944$_{0.91}$ | **-1.23$_{0.57}$** | **2.50$_{0.035}$** | 2.56$_{0.030}$ | 2.47$_{0.019}$ |
| SF2 | **-6.21$_{1.3}$** | -8.72$_{0.77}$ | 0.640$_{0.051}$ | **0.637$_{0.048}$** | 0.641$_{0.034}$ |
| SCP | **-5.17$_{2.2}$** | -7.58$_{0.51}$ | **0.392$_{0.099}$** | 0.398$_{0.099}$ | 0.400$_{0.069}$ |
| ANS | **1.89$_{0.025}$** | 1.91$_{0.017}$ | **0.531$_{0.0053}$** | 0.536$_{0.0049}$ | 0.532$_{0.0036}$ |
| HO2 | 3.06$_{0.052}$ | **2.87$_{0.029}$** | **0.881$_{0.0077}$** | 0.894$_{0.0068}$ | 0.881$_{0.0050}$ |
| SC2 | 1.88$_{0.25}$ | **0.936$_{0.12}$** | **1.01$_{0.0091}$** | 1.17$_{0.0096}$ | 1.09$_{0.0060}$ |
| RF1 | **-14.3$_{1.3}$** | -15.5$_{0.28}$ | **0.203$_{0.023}$** | 0.210$_{0.023}$ | 0.208$_{0.016}$ |
| SC1 | **-4.48$_{2.9}$** | -5.00$_{2.0}$ | 2.62$_{0.19}$ | **2.52$_{0.18}$** | 2.65$_{0.14}$ |
| AIR | 4.29$_{0.15}$ | **3.74$_{0.10}$** | **1.21$_{0.0097}$** | 1.23$_{0.0090}$ | 1.21$_{0.0065}$ |
| BI2 | -11.6$_{0.27}$ | **-12.3$_{0.16}$** | **0.831$_{0.018}$** | 0.839$_{0.017}$ | 0.830$_{0.011}$ |
| BI1 | **0.622$_{0.12}$** | 0.443$_{0.077}$ | **0.719$_{0.0049}$** | 0.722$_{0.0049}$ | 0.718$_{0.0034}$ |
| WAG | **-2.12$_{0.089}$** | -2.22$_{0.060}$ | 0.721$_{0.0045}$ | 0.716$_{0.0037}$ | **0.710$_{0.0027}$** |
| ME3 | -1.95$_{0.097}$ | **-2.37$_{0.049}$** | **0.401$_{0.0080}$** | 0.403$_{0.0078}$ | 0.398$_{0.0054}$ |
| ME1 | **-1.51$_{0.36}$** | -1.96$_{0.23}$ | 0.531$_{0.066}$ | **0.517$_{0.052}$** | 0.540$_{0.053}$ |
| ME2 | -1.94$_{0.076}$ | **-2.35$_{0.045}$** | **0.412$_{0.0047}$** | 0.412$_{0.0045}$ | 0.409$_{0.0033}$ |
| HO1 | -0.153$_{0.045}$ | **-0.323$_{0.023}$** | **0.327$_{0.0068}$** | 0.335$_{0.0070}$ | 0.330$_{0.0048}$ |
| BIO | -1.10$_{0.075}$ | **-1.42$_{0.016}$** | **0.203$_{0.0040}$** | 0.207$_{0.0040}$ | 0.204$_{0.0027}$ |
| BLO | **-3.37$_{0.35}$** | -3.85$_{0.24}$ | 0.372$_{0.015}$ | 0.372$_{0.018}$ | **0.362$_{0.0093}$** |
| CAL | **0.581$_{0.0083}$** | 0.578$_{0.0055}$ | **0.419$_{0.0014}$** | 0.421$_{0.0014}$ | 0.419$_{0.00094}$ |
| TAX | 1.62$_{0.0069}$ | **1.54$_{0.0042}$** | **0.696$_{0.0023}$** | 0.698$_{0.0021}$ | 0.695$_{0.0015}$ |

Table 13: Full comparative table, using a MAF model.

| | L-ECE | | HDR-ECE | | |
| --- | --- | --- | --- | --- | --- |
| | BASE | LR | BASE | HDR–R | LR |
| SLU | $0.146_{0.024}$ | $\mathbf{0.101_{0.013}}$ | $0.328_{0.026}$ | $\mathbf{0.110_{0.014}}$ | $0.277_{0.020}$ |
| EDM | $0.128_{0.020}$ | $\mathbf{0.0734_{0.0064}}$ | $0.153_{0.021}$ | $\mathbf{0.0849_{0.010}}$ | $0.0889_{0.013}$ |
| AT2 | $0.262_{0.019}$ | $\mathbf{0.0651_{0.0070}}$ | $0.376_{0.014}$ | $\mathbf{0.118_{0.016}}$ | $0.208_{0.010}$ |
| SF1 | $0.176_{0.0083}$ | $\mathbf{0.0487_{0.0044}}$ | $0.195_{0.013}$ | $0.129_{0.012}$ | $\mathbf{0.0935_{0.0062}}$ |
| OE2 | $0.300_{0.015}$ | $\mathbf{0.0595_{0.0048}}$ | $0.418_{0.011}$ | $\mathbf{0.201_{0.021}}$ | $0.220_{0.013}$ |
| AT1 | $0.258_{0.013}$ | $\mathbf{0.0566_{0.0069}}$ | $0.342_{0.011}$ | $\mathbf{0.106_{0.010}}$ | $0.159_{0.011}$ |
| JUR | $0.185_{0.013}$ | $\mathbf{0.0540_{0.0042}}$ | $0.314_{0.011}$ | $\mathbf{0.0717_{0.0094}}$ | $0.212_{0.010}$ |
| OE1 | $0.292_{0.013}$ | $\mathbf{0.0676_{0.0043}}$ | $0.391_{0.0097}$ | $0.174_{0.016}$ | $\mathbf{0.153_{0.015}}$ |
| ENB | $\mathbf{0.0277_{0.0044}}$ | $0.0346_{0.0035}$ | $\mathbf{0.0362_{0.0059}}$ | $0.0428_{0.0085}$ | $0.0461_{0.0060}$ |
| WQ | $0.328_{0.0095}$ | $\mathbf{0.0312_{0.0037}}$ | $0.349_{0.0067}$ | $\mathbf{0.134_{0.0083}}$ | $0.188_{0.0052}$ |
| SF2 | $0.174_{0.014}$ | $\mathbf{0.0310_{0.0026}}$ | $0.197_{0.018}$ | $0.123_{0.016}$ | $\mathbf{0.0726_{0.0042}}$ |
| SCP | $0.0885_{0.029}$ | $\mathbf{0.0397_{0.0038}}$ | $0.0813_{0.020}$ | $0.0445_{0.0052}$ | $\mathbf{0.0464_{0.0051}}$ |
| ANS | $\mathbf{0.0212_{0.0032}}$ | $0.0272_{0.0020}$ | $0.0443_{0.0055}$ | $0.0253_{0.0021}$ | $0.0478_{0.0036}$ |
| HO2 | $0.0806_{0.0038}$ | $\mathbf{0.0139_{0.00094}}$ | $0.131_{0.0043}$ | $\mathbf{0.0157_{0.0018}}$ | $0.0712_{0.0023}$ |
| SC2 | $0.250_{0.011}$ | $\mathbf{0.0166_{0.0015}}$ | $0.313_{0.0083}$ | $\mathbf{0.0672_{0.0061}}$ | $0.136_{0.0032}$ |
| RF1 | $0.0923_{0.0046}$ | $\mathbf{0.0172_{0.0038}}$ | $0.0860_{0.0062}$ | $\mathbf{0.0139_{0.0012}}$ | $0.0189_{0.0028}$ |
| SC1 | $0.112_{0.013}$ | $\mathbf{0.00913_{0.00061}}$ | $0.417_{0.017}$ | $\mathbf{0.163_{0.013}}$ | $0.394_{0.013}$ |
| AIR | $0.0936_{0.0028}$ | $\mathbf{0.0132_{0.0013}}$ | $0.144_{0.0049}$ | $\mathbf{0.0193_{0.0012}}$ | $0.0790_{0.0037}$ |
| BI2 | $0.0607_{0.0068}$ | $\mathbf{0.0122_{0.0013}}$ | $0.0885_{0.0062}$ | $\mathbf{0.0156_{0.0014}}$ | $0.0385_{0.0024}$ |
| BI1 | $0.0408_{0.0039}$ | $\mathbf{0.0105_{0.00081}}$ | $0.0739_{0.0064}$ | $\mathbf{0.0128_{0.00080}}$ | $0.0463_{0.0027}$ |
| WAG | $0.113_{0.0050}$ | $\mathbf{0.0118_{0.0010}}$ | $0.194_{0.0036}$ | $\mathbf{0.0134_{0.00090}}$ | $0.119_{0.0015}$ |
| ME3 | $0.0744_{0.0058}$ | $\mathbf{0.00796_{0.00065}}$ | $0.0919_{0.0046}$ | $\mathbf{0.0128_{0.0011}}$ | $0.0395_{0.0022}$ |
| ME1 | $0.0673_{0.013}$ | $\mathbf{0.0113_{0.00091}}$ | $0.0932_{0.016}$ | $\mathbf{0.0274_{0.0093}}$ | $0.0766_{0.016}$ |
| ME2 | $0.0716_{0.0057}$ | $\mathbf{0.00801_{0.00052}}$ | $0.0848_{0.0034}$ | $\mathbf{0.0135_{0.00087}}$ | $0.0333_{0.0018}$ |
| HO1 | $0.0843_{0.0019}$ | $\mathbf{0.00693_{0.00053}}$ | $0.0986_{0.0030}$ | $\mathbf{0.00896_{0.00069}}$ | $0.0292_{0.0014}$ |
| BIO | $0.0713_{0.0048}$ | $\mathbf{0.00615_{0.00056}}$ | $0.0850_{0.0054}$ | $\mathbf{0.00723_{0.00034}}$ | $0.0230_{0.0016}$ |
| BLO | $0.105_{0.019}$ | $\mathbf{0.00565_{0.00068}}$ | $0.102_{0.016}$ | $\mathbf{0.0152_{0.0026}}$ | $0.0285_{0.0034}$ |
| CAL | $\mathbf{0.00491_{0.00067}}$ | $0.00591_{0.00065}$ | $0.00543_{0.00065}$ | $0.00529_{0.00036}$ | $\mathbf{0.00416_{0.00027}}$ |
| TAX | $0.0235_{0.00098}$ | $\mathbf{0.00563_{0.00050}}$ | $0.0424_{0.0017}$ | $\mathbf{0.00797_{0.00075}}$ | $0.0235_{0.0012}$ |

# L   Results with Flow Matching

While our paper focuses on normalizing flows, LR is fully compatible with flow matching (FM) models (Section I.3), which also learn invertible mappings and assume a known latent distribution. For these models, we tune hyperparameters using grid search. The number of hidden units is chosen from $[32, 64]$, the number of hidden layers from $[2, 3, 5]$, and the learning rate from $[5 \times 10^{-3}, 10^{-3}, 2 \times 10^{-4}]$.

The FM results are aligned with the NFs results, with LR standing out particularly on the L-ECE and NLL metrics.

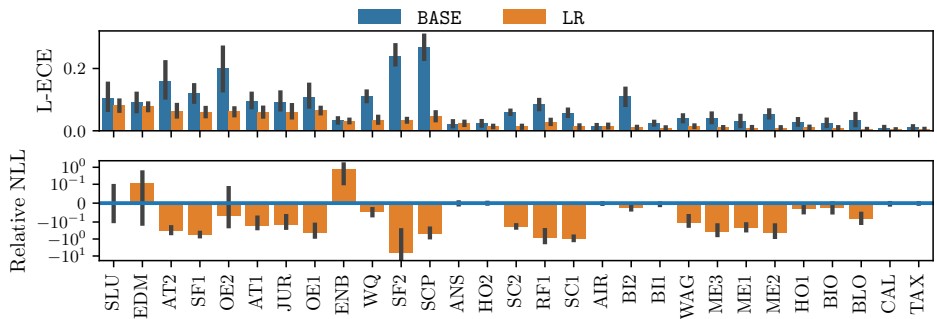

Figure 12: Latent calibration and NLL on datasets sorted by size, using a FM model.

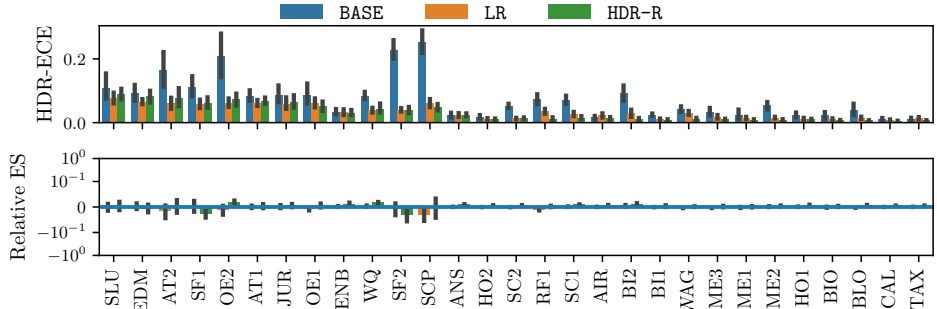

Figure 13: Latent calibration and NLL on datasets sorted by size, using a FM model.

Table 14: Full comparative table, using a FM model.

| | NLL | | Energy score | | |
|---|---|---|---|---|---|
| | BASE | LR | BASE | HDR-R | LR |
| SLU | $2.16_{0.20}$ | $2.09_{0.13}$ | $0.756_{0.033}$ | $0.759_{0.033}$ | $0.754_{0.029}$ |
| EDM | $2.20_{0.18}$ | $2.36_{0.35}$ | $0.642_{0.036}$ | $0.639_{0.037}$ | $0.643_{0.033}$ |
| AT2 | $4.76_{0.52}$ | $2.93_{0.21}$ | $0.770_{0.036}$ | $0.770_{0.032}$ | $0.756_{0.032}$ |
| SF1 | $2.38_{0.59}$ | $0.901_{0.17}$ | $0.845_{0.091}$ | $0.826_{0.092}$ | $0.841_{0.084}$ |
| OE2 | $9.53_{1.1}$ | $8.94_{1.4}$ | $1.20_{0.066}$ | $1.22_{0.066}$ | $1.18_{0.061}$ |
| AT1 | $1.46_{0.30}$ | $1.22_{0.24}$ | $0.571_{0.032}$ | $0.573_{0.032}$ | $0.571_{0.031}$ |
| JUR | $2.45_{0.25}$ | $2.02_{0.11}$ | $0.613_{0.027}$ | $0.615_{0.027}$ | $0.613_{0.026}$ |
| OE1 | $3.99_{1.4}$ | $2.24_{0.83}$ | $0.861_{0.057}$ | $0.866_{0.058}$ | $0.855_{0.057}$ |
| ENB | $0.139_{0.041}$ | $0.172_{0.035}$ | $0.313_{0.0042}$ | $0.316_{0.0047}$ | $0.313_{0.0041}$ |
| WQ | $15.0_{0.27}$ | $14.3_{0.25}$ | $2.40_{0.026}$ | $2.44_{0.023}$ | $2.41_{0.025}$ |
| SF2 | $2.30_{3.0}$ | $-1.06_{0.36}$ | $0.668_{0.043}$ | $0.648_{0.043}$ | $0.662_{0.041}$ |
| SCP | $-2.04_{0.59}$ | $-3.30_{0.18}$ | $0.391_{0.092}$ | $0.395_{0.097}$ | $0.386_{0.096}$ |
| ANS | $1.78_{0.023}$ | $1.77_{0.021}$ | $0.529_{0.0053}$ | $0.533_{0.0048}$ | $0.529_{0.0052}$ |
| HO2 | $2.53_{0.029}$ | $2.53_{0.029}$ | $0.852_{0.0069}$ | $0.856_{0.0066}$ | $0.852_{0.0068}$ |
| SC2 | $2.20_{0.17}$ | $1.82_{0.17}$ | $1.02_{0.0089}$ | $1.02_{0.0090}$ | $1.02_{0.0089}$ |
| RF1 | $0.570_{2.7}$ | $-4.53_{0.20}$ | $0.367_{0.018}$ | $0.367_{0.018}$ | $0.364_{0.018}$ |
| SC1 | $0.525_{0.14}$ | $0.0719_{0.13}$ | $0.818_{0.0077}$ | $0.825_{0.0078}$ | $0.819_{0.0078}$ |
| AIR | $4.21_{0.039}$ | $4.20_{0.040}$ | $1.16_{0.0090}$ | $1.17_{0.0089}$ | $1.16_{0.0090}$ |
| BI2 | $-4.20_{0.17}$ | $-4.31_{0.18}$ | $0.788_{0.013}$ | $0.796_{0.013}$ | $0.792_{0.013}$ |
| BI1 | $2.11_{0.013}$ | $2.10_{0.010}$ | $0.703_{0.0057}$ | $0.706_{0.0053}$ | $0.703_{0.0056}$ |
| WAG | $0.346_{0.036}$ | $0.308_{0.035}$ | $0.699_{0.0032}$ | $0.700_{0.0034}$ | $0.697_{0.0031}$ |
| ME3 | $-0.350_{0.073}$ | $-0.423_{0.069}$ | $0.392_{0.0087}$ | $0.393_{0.0090}$ | $0.391_{0.0086}$ |
| ME1 | $-0.444_{0.067}$ | $-0.518_{0.070}$ | $0.384_{0.0077}$ | $0.384_{0.0078}$ | $0.383_{0.0076}$ |
| ME2 | $-0.342_{0.058}$ | $-0.422_{0.054}$ | $0.395_{0.0045}$ | $0.397_{0.0048}$ | $0.396_{0.0045}$ |
| HO1 | $-0.619_{0.021}$ | $-0.636_{0.018}$ | $0.214_{0.0036}$ | $0.216_{0.0038}$ | $0.215_{0.0036}$ |
| BIO | $-0.561_{0.040}$ | $-0.570_{0.036}$ | $0.236_{0.0050}$ | $0.236_{0.0051}$ | $0.236_{0.0049}$ |
| BLO | $-1.06_{0.030}$ | $-1.14_{0.026}$ | $0.258_{0.0031}$ | $0.259_{0.0030}$ | $0.257_{0.0030}$ |
| CAL | $0.645_{0.0065}$ | $0.643_{0.0064}$ | $0.421_{0.0014}$ | $0.422_{0.0014}$ | $0.421_{0.0014}$ |
| TAX | $1.66_{0.0079}$ | $1.66_{0.0077}$ | $0.693_{0.0021}$ | $0.696_{0.0020}$ | $0.693_{0.0021}$ |

Table 15: Full comparative table, using a FM model.

| | L-ECE | | HDR-ECE | | |
| | BASE | LR | BASE | HDR-R | LR |
|---|---|---|---|---|---|
| SLU | $\mathbf{0.105_{0.022}}$ | $\mathbf{0.0809_{0.0088}}$ | $0.109_{0.021}$ | $0.0888_{0.0089}$ | $\mathbf{0.0761_{0.0092}}$ |
| EDM | $\mathbf{0.0906_{0.015}}$ | $\mathbf{0.0776_{0.0057}}$ | $0.0926_{0.014}$ | $0.0820_{0.0097}$ | $\mathbf{0.0654_{0.0042}}$ |
| AT2 | $0.160_{0.030}$ | $\mathbf{0.0621_{0.010}}$ | $0.165_{0.030}$ | $0.0784_{0.016}$ | $\mathbf{0.0605_{0.0097}}$ |
| SF1 | $0.119_{0.015}$ | $\mathbf{0.0593_{0.0072}}$ | $0.112_{0.017}$ | $0.0604_{0.0090}$ | $\mathbf{0.0572_{0.0078}}$ |
| OE2 | $0.199_{0.036}$ | $\mathbf{0.0615_{0.0063}}$ | $0.210_{0.036}$ | $0.0722_{0.0098}$ | $\mathbf{0.0626_{0.0072}}$ |
| AT1 | $0.0959_{0.012}$ | $\mathbf{0.0597_{0.0074}}$ | $0.0831_{0.0089}$ | $0.0679_{0.0057}$ | $\mathbf{0.0615_{0.0051}}$ |
| JUR | $0.0918_{0.015}$ | $\mathbf{0.0596_{0.011}}$ | $\mathbf{0.0865_{0.014}}$ | $0.0639_{0.011}$ | $\mathbf{0.0590_{0.010}}$ |
| OE1 | $0.109_{0.019}$ | $\mathbf{0.0647_{0.0049}}$ | $0.0863_{0.017}$ | $0.0514_{0.0078}$ | $0.0618_{0.0077}$ |
| ENB | $\mathbf{0.0330_{0.0037}}$ | $\mathbf{0.0314_{0.0021}}$ | $\mathbf{0.0340_{0.0044}}$ | $0.0312_{0.0042}$ | $0.0318_{0.0048}$ |
| WQ | $0.110_{0.0086}$ | $\mathbf{0.0349_{0.0049}}$ | $0.0842_{0.0061}$ | $0.0437_{0.0078}$ | $0.0385_{0.0042}$ |
| SF2 | $0.239_{0.016}$ | $\mathbf{0.0339_{0.0023}}$ | $0.226_{0.015}$ | $0.0397_{0.0051}$ | $0.0400_{0.0030}$ |
| SCP | $0.268_{0.020}$ | $\mathbf{0.0457_{0.0070}}$ | $0.253_{0.019}$ | $0.0483_{0.0053}$ | $0.0607_{0.0067}$ |
| ANS | $\mathbf{0.0222_{0.0042}}$ | $\mathbf{0.0236_{0.0027}}$ | $\mathbf{0.0231_{0.0040}}$ | $0.0244_{0.0025}$ | $0.0236_{0.0033}$ |
| HO2 | $0.0232_{0.0038}$ | $\mathbf{0.0138_{0.0011}}$ | $0.0180_{0.0028}$ | $0.0109_{0.0011}$ | $0.0116_{0.0013}$ |
| SC2 | $0.0590_{0.0028}$ | $\mathbf{0.0129_{0.0017}}$ | $0.0532_{0.0032}$ | $0.0142_{0.0011}$ | $0.0127_{0.0012}$ |
| RF1 | $0.0843_{0.0078}$ | $\mathbf{0.0273_{0.0037}}$ | $0.0740_{0.0082}$ | $0.0125_{0.0021}$ | $0.0353_{0.0039}$ |
| SC1 | $0.0573_{0.0056}$ | $\mathbf{0.0133_{0.0020}}$ | $0.0720_{0.0066}$ | $0.0156_{0.0021}$ | $0.0267_{0.0037}$ |
| AIR | $\mathbf{0.0130_{0.0027}}$ | $\mathbf{0.0151_{0.0023}}$ | $\mathbf{0.0158_{0.0027}}$ | $0.0132_{0.0021}$ | $0.0220_{0.0035}$ |
| BI2 | $0.109_{0.014}$ | $\mathbf{0.00979_{0.0014}}$ | $0.0916_{0.013}$ | $0.0112_{0.0016}$ | $0.0288_{0.0059}$ |
| BI1 | $0.0243_{0.0023}$ | $\mathbf{0.00924_{0.00093}}$ | $0.0249_{0.0020}$ | $\mathbf{0.00927_{0.00061}}$ | $\mathbf{0.0101_{0.0013}}$ |
| WAG | $0.0407_{0.0051}$ | $\mathbf{0.0144_{0.0013}}$ | $0.0419_{0.0050}$ | $0.0120_{0.0014}$ | $0.0307_{0.0031}$ |
| ME3 | $0.0412_{0.0077}$ | $\mathbf{0.00965_{0.0011}}$ | $0.0330_{0.0065}$ | $0.0104_{0.00098}$ | $0.0176_{0.0029}$ |
| ME1 | $0.0299_{0.0086}$ | $\mathbf{0.00915_{0.0014}}$ | $0.0251_{0.0075}$ | $\mathbf{0.00935_{0.00058}}$ | $0.0143_{0.0020}$ |
| ME2 | $0.0537_{0.0060}$ | $\mathbf{0.00883_{0.0013}}$ | $0.0541_{0.0063}$ | $\mathbf{0.00892_{0.00094}}$ | $0.0156_{0.0014}$ |
| HO1 | $0.0285_{0.0046}$ | $\mathbf{0.00990_{0.0016}}$ | $0.0239_{0.0043}$ | $\mathbf{0.00974_{0.0011}}$ | $\mathbf{0.0119_{0.0016}}$ |
| BIO | $0.0253_{0.0053}$ | $\mathbf{0.00766_{0.0019}}$ | $0.0221_{0.0058}$ | $\mathbf{0.00726_{0.00062}}$ | $0.0102_{0.0016}$ |
| BLO | $0.0346_{0.010}$ | $\mathbf{0.00498_{0.00047}}$ | $0.0402_{0.0099}$ | $\mathbf{0.00648_{0.00050}}$ | $0.0153_{0.0018}$ |
| CAL | $0.00951_{0.0014}$ | $\mathbf{0.00395_{0.00043}}$ | $0.0112_{0.0018}$ | $\mathbf{0.00547_{0.00030}}$ | $0.00746_{0.0011}$ |
| TAX | $0.0110_{0.0018}$ | $\mathbf{0.00505_{0.00059}}$ | $0.0122_{0.0017}$ | $\mathbf{0.00660_{0.00054}}$ | $0.0133_{0.0022}$ |

# M    Results with a misspecified convex potential flow

Tables 16 and 17 along with Figure 14 and Figure 15 present results for a deliberately misspecified convex potential flow. This misspecification was induced by training the base predictor for only two epochs, ensuring it has low predictive accuracy and is likely poorly calibrated.

We observe that, in this additional scenario, LR also leads to improved L-ECE and NLL on most datasets, indicating enhanced predictive accuracy compared to the BASE misspecified model. LR achieves similar or improved HDR-ECE and ES compared to HDR-R. These results highlight LR's ability to improve misspecified base predictors.

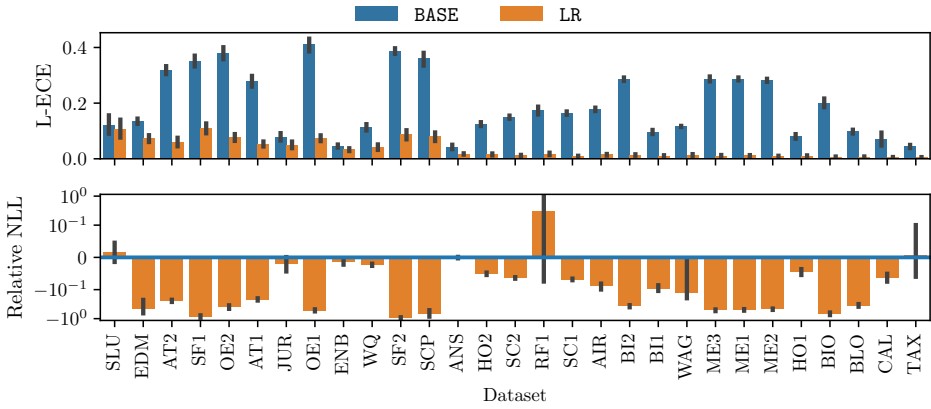

Figure 14: Latent calibration and NLL on datasets sorted by size, using a misspecified convex potential flow model.

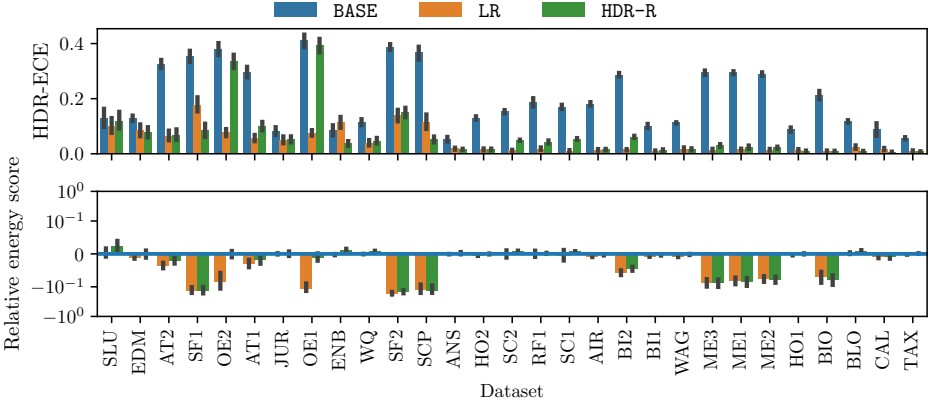

Figure 15: Latent calibration and NLL on datasets sorted by size, using a misspecified convex potential flow model.

Table 16: Full comparative table, using a misspecified convex potential flow model.

| | NLL | | Energy score | | |
| | BASE | LR | BASE | HDR–R | LR |
|---|---|---|---|---|---|
| SLU | $3.86_{0.13}$ | $3.93_{0.16}$ | $1.01_{0.043}$ | $1.03_{0.050}$ | $1.01_{0.047}$ |
| EDM | $2.84_{0.059}$ | $1.53_{0.32}$ | $0.880_{0.028}$ | $0.879_{0.029}$ | $0.871_{0.029}$ |
| AT2 | $7.99_{0.29}$ | $6.03_{0.14}$ | $1.39_{0.051}$ | $1.36_{0.051}$ | $1.34_{0.054}$ |
| SF1 | $4.05_{0.39}$ | $0.736_{0.22}$ | $0.857_{0.075}$ | $0.754_{0.083}$ | $0.753_{0.079}$ |
| OE2 | $23.1_{2.2}$ | $13.5_{0.80}$ | $2.24_{0.14}$ | $2.24_{0.15}$ | $2.07_{0.16}$ |
| AT1 | $7.68_{0.38}$ | $5.99_{0.26}$ | $1.42_{0.070}$ | $1.40_{0.078}$ | $1.39_{0.076}$ |
| JUR | $4.19_{0.13}$ | $4.09_{0.075}$ | $1.08_{0.034}$ | $1.08_{0.036}$ | $1.08_{0.034}$ |
| OE1 | $23.8_{2.1}$ | $11.1_{0.65}$ | $2.14_{0.13}$ | $2.12_{0.14}$ | $1.91_{0.15}$ |
| ENB | $2.05_{0.093}$ | $2.02_{0.090}$ | $0.815_{0.020}$ | $0.823_{0.020}$ | $0.814_{0.019}$ |
| WQ | $20.0_{0.15}$ | $19.6_{0.15}$ | $2.59_{0.027}$ | $2.61_{0.028}$ | $2.59_{0.027}$ |
| SF2 | $4.71_{0.55}$ | $0.298_{0.25}$ | $0.821_{0.038}$ | $0.703_{0.039}$ | $0.688_{0.038}$ |
| SCP | $5.06_{2.1}$ | $1.25_{0.36}$ | $0.738_{0.10}$ | $0.657_{0.11}$ | $0.658_{0.11}$ |
| ANS | $2.61_{0.021}$ | $2.61_{0.020}$ | $0.817_{0.0089}$ | $0.819_{0.0087}$ | $0.817_{0.0090}$ |
| HO2 | $5.46_{0.020}$ | $5.18_{0.021}$ | $1.23_{0.0063}$ | $1.23_{0.0070}$ | $1.22_{0.0067}$ |
| SC2 | $22.2_{0.12}$ | $20.8_{0.12}$ | $2.67_{0.019}$ | $2.69_{0.020}$ | $2.66_{0.021}$ |
| RF1 | $10.8_{0.84}$ | $16.8_{7.5}$ | $1.69_{0.026}$ | $1.70_{0.026}$ | $1.69_{0.032}$ |
| SC1 | $21.4_{0.099}$ | $20.0_{0.10}$ | $2.56_{0.023}$ | $2.58_{0.024}$ | $2.54_{0.023}$ |
| BI2 | $5.62_{0.10}$ | $3.51_{0.085}$ | $1.08_{0.016}$ | $1.03_{0.016}$ | $1.02_{0.016}$ |
| BI1 | $2.66_{0.030}$ | $2.39_{0.020}$ | $0.779_{0.0057}$ | $0.776_{0.0057}$ | $0.773_{0.0055}$ |
| AIR | $8.33_{0.068}$ | $7.58_{0.044}$ | $1.50_{0.0089}$ | $1.50_{0.0089}$ | $1.49_{0.0090}$ |
| WAG | $2.78_{0.0069}$ | $2.43_{0.17}$ | $0.876_{0.0029}$ | $0.875_{0.0027}$ | $0.868_{0.0029}$ |
| ME3 | $2.31_{0.045}$ | $1.12_{0.046}$ | $0.602_{0.010}$ | $0.548_{0.0087}$ | $0.549_{0.0091}$ |
| ME1 | $2.20_{0.046}$ | $1.10_{0.048}$ | $0.588_{0.0092}$ | $0.539_{0.0073}$ | $0.540_{0.0073}$ |
| ME2 | $2.15_{0.027}$ | $1.13_{0.038}$ | $0.591_{0.0055}$ | $0.545_{0.0059}$ | $0.546_{0.0058}$ |
| HO1 | $2.46_{0.029}$ | $2.34_{0.030}$ | $0.743_{0.0068}$ | $0.743_{0.0072}$ | $0.741_{0.0071}$ |
| BIO | $0.963_{0.14}$ | $0.302_{0.046}$ | $0.335_{0.0085}$ | $0.308_{0.0058}$ | $0.311_{0.0062}$ |
| CAL | $1.37_{0.043}$ | $1.28_{0.035}$ | $0.472_{0.0061}$ | $0.468_{0.0050}$ | $0.468_{0.0052}$ |
| BLO | $1.21_{0.033}$ | $0.785_{0.036}$ | $0.685_{0.0024}$ | $0.691_{0.0030}$ | $0.687_{0.0029}$ |
| TAX | $2.29_{0.019}$ | $2.31_{0.11}$ | $0.724_{0.0024}$ | $0.725_{0.0024}$ | $0.723_{0.0024}$ |

Table 17: Full comparative table, using a misspecified convex potential flow model.

| | L-ECE | | HDR-ECE | | |
| --- | --- | --- | --- | --- | --- |
| | BASE | LR | BASE | HDR–R | LR |
| SLU | **$0.120_{0.018}$** | **$0.107_{0.017}$** | **$0.127_{0.018}$** | **$0.119_{0.017}$** | **$0.101_{0.015}$** |
| EDM | $0.135_{0.0052}$ | **$0.0734_{0.0068}$** | $0.128_{0.0051}$ | **$0.0778_{0.0097}$** | $0.0846_{0.012}$ |
| AT2 | $0.319_{0.0075}$ | **$0.0593_{0.0083}$** | $0.327_{0.0078}$ | **$0.0673_{0.011}$** | **$0.0642_{0.0095}$** |
| SF1 | $0.351_{0.011}$ | **$0.110_{0.0094}$** | $0.354_{0.011}$ | **$0.0838_{0.012}$** | $0.177_{0.014}$ |
| OE2 | $0.379_{0.012}$ | **$0.0773_{0.0067}$** | $0.379_{0.012}$ | $0.335_{0.013}$ | **$0.0769_{0.0071}$** |
| AT1 | $0.278_{0.010}$ | **$0.0533_{0.0051}$** | $0.297_{0.011}$ | $0.100_{0.0080}$ | **$0.0565_{0.0063}$** |
| JUR | $0.0785_{0.0070}$ | **$0.0494_{0.0064}$** | $0.0819_{0.0078}$ | **$0.0518_{0.0062}$** | **$0.0501_{0.0068}$** |
| OE1 | $0.410_{0.013}$ | **$0.0733_{0.0057}$** | $0.410_{0.013}$ | $0.395_{0.013}$ | **$0.0749_{0.0054}$** |
| ENB | $0.0458_{0.0030}$ | **$0.0326_{0.0028}$** | $0.0854_{0.0099}$ | **$0.0367_{0.0049}$** | $0.114_{0.010}$ |
| WQ | $0.112_{0.0060}$ | **$0.0405_{0.0054}$** | $0.114_{0.0060}$ | **$0.0467_{0.0051}$** | **$0.0391_{0.0053}$** |
| SF2 | $0.386_{0.0057}$ | **$0.0869_{0.0087}$** | $0.387_{0.0057}$ | **$0.150_{0.0090}$** | **$0.139_{0.011}$** |
| SCP | $0.361_{0.013}$ | **$0.0808_{0.0082}$** | $0.369_{0.013}$ | **$0.0517_{0.0063}$** | $0.115_{0.014}$ |
| ANS | $0.0428_{0.0043}$ | **$0.0172_{0.0012}$** | $0.0524_{0.0047}$ | **$0.0162_{0.0011}$** | **$0.0183_{0.0019}$** |
| HO2 | $0.125_{0.0038}$ | **$0.0155_{0.0019}$** | $0.130_{0.0035}$ | **$0.0155_{0.0016}$** | **$0.0156_{0.0020}$** |
| SC2 | $0.150_{0.0030}$ | **$0.0123_{0.00093}$** | $0.153_{0.0032}$ | $0.0491_{0.00096}$ | **$0.0124_{0.00095}$** |
| RF1 | $0.173_{0.0076}$ | **$0.0172_{0.0024}$** | $0.186_{0.0081}$ | $0.0422_{0.0030}$ | **$0.0180_{0.0025}$** |
| SC1 | $0.164_{0.0033}$ | **$0.00888_{0.0010}$** | $0.170_{0.0040}$ | $0.0541_{0.0016}$ | **$0.0103_{0.0014}$** |
| BI2 | $0.286_{0.0034}$ | **$0.0133_{0.0014}$** | $0.287_{0.0034}$ | $0.0616_{0.0019}$ | **$0.0146_{0.0013}$** |
| BI1 | $0.0961_{0.0039}$ | **$0.00962_{0.0014}$** | $0.100_{0.0040}$ | $0.0126_{0.0016}$ | **$0.0102_{0.00095}$** |
| AIR | $0.178_{0.0034}$ | **$0.0146_{0.0016}$** | $0.181_{0.0033}$ | **$0.0151_{0.0013}$** | **$0.0140_{0.0016}$** |
| WAG | $0.116_{0.0011}$ | **$0.0127_{0.0021}$** | $0.113_{0.0011}$ | $0.0158_{0.0019}$ | **$0.0166_{0.0033}$** |
| ME3 | $0.287_{0.0047}$ | **$0.0105_{0.0016}$** | $0.294_{0.0040}$ | $0.0304_{0.0025}$ | **$0.0140_{0.0022}$** |
| ME1 | $0.287_{0.0029}$ | **$0.0110_{0.0011}$** | $0.295_{0.0027}$ | $0.0237_{0.0031}$ | **$0.0149_{0.0020}$** |
| ME2 | $0.282_{0.0031}$ | **$0.00893_{0.00095}$** | $0.290_{0.0034}$ | $0.0220_{0.0021}$ | **$0.0144_{0.0017}$** |
| HO1 | $0.0801_{0.0046}$ | **$0.0102_{0.0012}$** | $0.0880_{0.0041}$ | **$0.00989_{0.00082}$** | $0.0120_{0.0024}$ |
| BIO | $0.200_{0.0086}$ | **$0.00695_{0.00065}$** | $0.213_{0.0080}$ | **$0.00985_{0.00048}$** | **$0.00916_{0.0013}$** |
| CAL | $0.0711_{0.013}$ | **$0.00549_{0.00046}$** | $0.0883_{0.013}$ | **$0.00681_{0.00041}$** | $0.0169_{0.0024}$ |
| BLO | $0.0972_{0.0039}$ | **$0.00657_{0.00092}$** | $0.117_{0.025}$ | **$0.00843_{0.00092}$** | $0.0241_{0.0033}$ |
| TAX | $0.0444_{0.0032}$ | **$0.00490_{0.00060}$** | $0.0565_{0.0020}$ | **$0.00804_{0.00053}$** | **$0.00918_{0.0015}$** |

