# OpenReview forum: "Multivariate Latent Recalibration for Conditional Normalizing Flows"
_NeurIPS.cc/2025/Conference — NeurIPS 2025 poster_

### Official Review · Reviewer_54XG · 2025-06-12

**Clarity:** 2
**Significance:** 2
**Originality:** 2
**Rating:** 4
**Confidence:** 2

**Summary:**

This paper proposes latent calibration and latent recalibration (LR) for calibrating multivariate predictive distributions in conditional normalizing flows. Latent calibration is formalized through the probability integral transform (PIT) of the norm of the latent vectors. LR is a post-hoc procedure, which learns a transformation in the latent space and outputs an explicit probability density. In experiments on tabular and image datasets, the proposed LR method is empirically shown to be effective in increasing predictive accuracy and reducing calibration errors in terms of L-ECE and HDR-ECE.

**Questions:**

- I personally don't get what the black bars in Figure 2 and the green bars in Figure 3 mean. Perhaps the green bars are HDR-R. Can the authors clarify these two figures more?
- For the image task, can the authors provide some qualitative inspections and analysis of how LR performs?

**Ethical Concerns:**

["NO or VERY MINOR ethics concerns only"]

**Final Justification:**

The additional clarifications and the new experiments the authors have conducted are helpful. My initial concerns about claims on computational efficiencies have been resolved by new experimental results. I increase my score to 4.

**Limitations:**

Yes.

**Paper Formatting Concerns:**

I didn't notice any.

**Quality:**

3

**Strengths And Weaknesses:**

Strengths:
- The paper is well organized.
- The empirical evaluations are extensive, considering a large set of tabular datasets and a high-dimensional image dataset.


Weaknesses:
- It seems LR relies on invertibility and the latent (norm) distribution needs to be known, which limits the extension in other generative models (e.g. VAEs and diffusion models).
- The authors claim that LR is computationally efficient, which is not backed up by experiments.
- It seems there are no comparisons against other recalibration methods in experiments.

---

> ### Author Rebuttal · Authors · 2025-07-31
>
> Thank you for the valuable feedback and for recognizing the extensiveness of our experiments.
>
> **On the extension to other models**
>
> We discuss this point in detail in our response to Reviewer 4tSF. While `LR` is not applicable to VAEs and standard diffusion models, it is directly extendable to **Conditional Flow Matching (CFM)**. We showed that `LR` also leads to significantly improved performance with CFM.
>
> **On the comparison to other methods**
>
> We compared `LR` to the uncalibrated baseline and `HDR-R` because, to our knowledge, these are the only existing methods for post-hoc recalibration of multi-dimensional distributions. The authors of `HDR-R` [1] also noted that "methods which account for the joint multi-dimensional distribution in assessing calibration and recalibrating the prediction are generally lacking." Our work helps fill this important gap.
>
> [1] Chung et al (2024). Sampling-based Multi-dimensional Recalibration. ICML.
>
> **Computational efficiency**
>
> We agree that reporting the computation time will strengthen the paper.
> The difference in computation time can be measured in two aspects:
> - For calibration, the computational complexity of `HDR-R` is $O(M F n)$ and `LR` is $O(R n)$ where $M = 100$ corresponds to the number of samples of `HDR-R` per instance, $F$ the time for the forward mapping $\hat{T}$ and $R$ the time for the reverse mapping $\hat{T}^{-1}$.
> - For inference, it is a bit more subtle. Given a test insance $x$, `HDR-R` requires to sample at least $M$ times ($O(M F)$) to obtain a recalibrated sample, which can be a weakness, e.g., if only one conditional sample is needed. `LR` only incurs a low fixed cost $C$ for evaluating the recalibration map ($O(C + F)$). Thus, the inference time is not directly comparable.
>
> We report the calibration time of `HDR-R` and `LR` in seconds, using the convex potential flow model and averaged over 10 runs. To avoid the table being too large, we restrict the table to the largest datasets.
>
> | Method |      HO2 |      SC2 |      RF1 |     SC1 |      AIR |       BI2 |      BI1 |       WAG |      ME3 |       ME1 |       ME2 |      HO1 |       BIO |        BLO |     CAL |       TAX |
> |:-----|---------:|---------:|---------:|--------:|---------:|----------:|---------:|----------:|---------:|----------:|----------:|---------:|----------:|-----------:|--------:|----------:|
> | `HDR-R` | 1.56     | 2.78     | 4.9      | 2.57    | 4.86     | 18.8      | 8.8      | 15        | 9.98     | 12.7      | 19.3      | 7.19     | 37.7      | 168        | 8.53    | 10.5      |
> | `LR`    | 0.232    | 0.185    | 0.156    | 0.187   | 0.149    |  0.142    | 0.133    |  0.133    | 0.152    |  0.153    |  0.163    | 0.192    |  0.29     |   0.328    | 0.482   |  0.324    |
>
> On CIFAR-10 with TarFlow, the time difference is larger and can be prohibitive for `HDR-R`:
>
> | Method | CIFAR-10 |
> |:---|---:|
> | `HDR-R` | 183182 |
> | `LR`    | 1259 |
>
> **Clarifications for the figures**
>
> Thank you for pointing out the ambiguity in our figures. You are correct that the green bars in Figure 3 represent the `HDR-R` method. The bars in Figures 2 and 3 show the mean metric value across 10 independent runs, and the black error bars depict one standard error of the mean. We apologize for the omission and will update the figure captions and legends to make this explicit.
>
> **Image generation quality**
>
> This is a great question. We clarify this point in detail in our answer to Reviewer qdRU.

---

> > ### Author Response · Authors · 2025-08-05
> >
> > Dear Reviewer 54XG,
> >
> > Thank you again for your time reviewing our manuscript. We hope that our rebuttal has addressed your concerns.
> >
> > We would greatly appreciate if you could let us know if any points remain unclear or if you have any further feedback before the discussion period ends.
> >
> > Best regards,
> >
> > The Authors.

---

> > ### Comment · Reviewer_54XG · 2025-08-05
> >
> > Thank you for the additional clarifications and the new experiments you have conducted. I have also read the other reviews and rebuttals. While some of the points raised were new to me and quite insightful, I believe this work demonstrates promising extensibility beyond normalizing flows, as well as meaningful computational advantages, now supported by additional empirical results. I will update my score accordingly.

---

### Official Review · Reviewer_qdRU · 2025-06-28

**Clarity:** 3
**Significance:** 3
**Originality:** 3
**Rating:** 4
**Confidence:** 3

**Summary:**

This paper propose a new notion "Latent Calibration" for multivariate calibration, which aims to overcome the shortcomings of HDR-R, which calibrates the highest-density regtions (HDR). Extensive experiments are conducted to demonstrate the usefulness of the proposed method "LR", and it has been demonstrated that 1) the NLL can be improved with LR, 2) LR performs the best when evaluating with the L-ECE.

**Questions:**

I have the following minor questions/suggestions:
- In table 4, I find it quite surprising that the energy score for BASE seems to be very close to LR and HDR-R. Do you have any intuitions why this happened?

- You've already included the image dataset in your experiments. Do you observe any "quality" difference when you use HDR-R/LR to calibrate NF and then to generate new samples?

**Ethical Concerns:**

["NO or VERY MINOR ethics concerns only"]

**Final Justification:**

Strength:
- Having explicit pdf is a hard task, and the proposed method is nice and novel
- The paper is well-written and very easy to follow.
- The scale of the experiments is quite large.

Weakness:
- The experiment results on synthetic datasets are quite strong, yet those on the real-world datasets are less convincing.

I hence think this is a boardline paper (towards acceptance).

**Limitations:**

Yes

**Quality:**

2

**Strengths And Weaknesses:**

Strength:
- The paper is in genereal well-written and very easy to follow.
- The scale of the experiments is quite large.
- Multivariate calibration is a hard problem in general, and having explicit PDF is a nice property in my opinion.

Weakness:
- The target of the paper is a bit vague to me. It's not clear if the primary goal is 1) to promote the latent calibration over HDR calibration as a task/research question, 2) to propose an alternative multivariate calibration method that is faster and has explicit PDF.

- The paper does not provide sufficient evidence to either of the two targets above. Specifically:

a) For target (1), some theoretic work or empirical experiments might be needed to show the advantage of latent calibration over HDR calibration; e.g., a synthetic case for which HDR calibration fails but latent calibration performs well, or LR improves the quality of generative samples of that of HDR in some cases, or a case study to demonstrate the usefulness of having explicit PDF.

b) For target (2), the motivation of having explicit PDFs is lacking in the introduction; further, the experiment to demonstrate the runtime is missing.

---

> ### Author Rebuttal · Authors · 2025-07-31
>
> Thank you for your valuable feedback and recognizing the paper's clarity and the experimental evaluation.
>
> **Clarifying the paper's goal and contributions**
>
> Our main contribution is to introduce latent calibration, a principled framework for multivariate calibration, and simultaneously propose Latent Recalibration (`LR`), a practical and efficient method that realizes this framework. The two are intrinsically linked: the framework provides the theoretical foundation, and the method demonstrates its practical value.
>
> A key advantage of our approach is that `LR` produces a fully-specified, recalibrated PDF. This is a critical feature that distinguishes it from set-based or sampling-based methods and enables superior performance in important downstream tasks, as we demonstrate below.
>
> **1) Performance of LR on a synthetic experiment**
>
> To provide further evidence of the improved performance of `LR`, we compare the performance of `LR` and `HDR-R` on the synthetic dataset of Figure 1.
> The `BASE` model is a Masked Autoregressive Flow that has been misspecified by being only trained for 5 epochs.
> The dataset is divided into 2500 training points, 500 validation/calibration points and 2000 test points.
> The metrics, averaged over 200 runs, are displayed below with the standard error of the mean.
>
> | NLL `BASE` | NLL `LR` | ES `BASE` | ES `HDR-R` | ES `LR` |
> |:---|:---|:---|:---|:---|
> | 1.83 ± 0.022 | **1.29 ± 0.019** | 0.930 ± 0.0029 | 0.922 ± 0.0027 | **0.881 ± 0.0012** |
>
> `LR` significantly improves both NLL and the Energy Score (ES). `HDR-R`, which is limited to ES evaluation, shows a much smaller improvement. This highlights that `LR` can correct model deficiencies that `HDR-R` does not.
>
> **2) Practical advantages of producing an explicit PDF**
>
> This is an excellent point. To make the benefits of a full PDF concrete, we have conducted a new experiment on a decision-making task, which we will add to the paper.
>
> **Motivation:** Access to a full, calibrated PDF is crucial for any task requiring estimation of the probability mass within an **arbitrary, non-standard region** of the output space, a capability that set-based methods (like Conformal Prediction) or pure sampling-based methods do not provide. A direct application is anomaly detection, where low-density points are classified as anomaly [1, 2]. Other examples include risk assessment in engineering, targeted material design, or optimal control.
>
> [1] Amit Rozner et al (2024). Anomaly Detection with Variance Stabilized Density Estimation. UAI.
>
> [2] Lorenzo Perini et al (2024). Uncertainty-aware Evaluation of Auxiliary Anomalies with the Expected Anomaly Posterior. TMLR.
>
> **Experimental Setup:** We use the SLUMP dataset (first dataset in Figure 3), where inputs are ingredients for producing concrete and outputs $Y = (S, F, C) \in \mathbb{R}^3$ are three concrete properties. A manufacturer must decide whether a given batch of ingredients is suitable for one of two projects (A or B), each with specific, non-negotiable requirement regions, or if it should be discarded. The decision has different financial utilities and risks.
> *   *Project A requirements*: $7 \leq S \leq 20, 55 \leq F \leq 65, 25 \leq C \leq 40$.
> *   *Project B requirements*: $20 \leq S \leq 29, 70 \leq F \leq 100, 15 \leq C \leq 30$.
> *   The optimal action is chosen by maximizing the expected utility, which depends on the estimated probabilities $\hat{P}(Y \in \text{Region}_A | X)$ and $\hat{P}(Y \in \text{Region}_B | X)$. These probabilities are estimated either by (1) sampling or (2) numerical integration of the PDF.
> We use either 125 samples for sampling or a 5 x 5 x 5 grid for integration using trapezoidal rule.
>
> **Results:**
> | Method | Estimation Strategy | Average Utility |
> | :--- | :--- | :--- |
> | `BASE` | Sampling | 62.53 ± 11.33 |
> | `HDR-R` | Sampling | 32.38 ± 10.81 |
> | `BASE` | PDF (Numerical Integration) | 76.23 ± 11.99 |
> | **LR** | **PDF (Numerical Integration)** | **113.31 ± 12.91** |
>
> The results show two key observations:
> 1.  Using the PDF via numerical integration leads to better decisions (higher utility) than relying on a finite number of samples.
> 2.  The improved calibration from `LR` provides a more accurate PDF, leading to a significant further increase in utility. `HDR-R`, which relies on resampling from the original uncalibrated density, actually harmed decision quality in this task.
>
> This demonstrates a concrete scenario where an explicit, calibrated PDF is practically useful for optimal decision-making.
>
> **3) Computational efficiency**
>
> You are correct that we should substantiate our claim of computational efficiency with experimental results. As detailed in our response to Reviewer 54XG, we have now benchmarked the calibration time of `LR` against `HDR-R`. The results confirm that `LR` is substantially faster, particularly on larger datasets and computationally intensive models.
>
> **Energy score**
>
> Thank you for the insightful question, which touches upon a point we briefly noted in Section 5.3 regarding the Energy Score's (ES) weaker discriminative ability. The observation that the ES remains largely unchanged is an expected outcome that stems from the score's fundamental limitations in discriminative ability. As established in [1] and corroborated by [2], the ES is sensitive to shifts in the mean but notoriously insensitive to misspecifications in variance, correlation, and overall dependency structure. `LR` is a post-hoc procedure that primarily corrects the shape and spread of the predictive distribution. Therefore, the ES is fundamentally ill-suited to capture the specific improvements `LR` provides.
>
> In contrast, the NLL is uniquely suited for this evaluation. As the only local strictly proper scoring rule, its value depends only on the probability density at the precise location of the observed outcome [3]. This locality makes it highly discerning of the very improvements `LR` makes to the distributional shape, which is why we observe significant and consistent NLL reductions. This result thus highlights an inherent property of the evaluation metric itself, not a shortcoming of our method, and we will clarify this important distinction in the revised manuscript.
>
> [1] Pinson, P., & Tastu, J. (2013). Discrimination ability of the energy score. Technical Report.
>
> [2] Alexander, C., et al (2022). Evaluating the discrimination ability of proper multi-variate scoring rules. Annals of Operations Research.
>
> [3] Du, H. (2021). Beyond Strictly Proper Scoring Rules: The Importance of Being Local. Weather and Forecasting.
>
> **Image generation quality**
>
> This is a great question, and we appreciate the opportunity to clarify this point. To be perfectly clear, `LR` is not expected to improve the visual quality of generated samples. We will state this explicitly in the paper to avoid any potential confusion. The goal of our image data experiment is to understand the behaviour of latent calibration and recalibration in a very-high dimensional setting.
>
> In the noisy setting, `LR` preserves the visual quality of the samples from the base model, with no perceptually visible changes, which aligns with the very small change we observed in NLL. In the noiseless setting, where the test data is out-of-distribution relative to the noisy training data, applying `LR` can cause samples to appear less realistic with more uniform colors. This is an expected outcome, as a simple post-hoc recalibration step cannot be expected to generalize to a domain shift and improve generation quality.
>
> Samples from `HDR-R` are only resampled from an initial pool of samples. Thus, differences in visual quality are not expected either.

---

> > ### Comment · Reviewer_qdRU · 2025-08-04
> >
> > Dear authors,
> >
> > Thanks a lot. Although I do agree that the results on synthetic datasets (fig 1) can provide additional insights on the empirical performance of the proposed method, I don't find the experiment results only on synthetic datasets can change my view on the results of table 4 as table 4 is about "real-world" tabular datasets.
> >
> > As said in my initial review, I really like the property of explicity pdf, although I do think it is not emphasized enough in the manuscript. A case study like what you have in the rebuttal would definitely be helpful.

---

> ### Author Response · Authors · 2025-08-05
>
> Dear Reviewer qdRU,
>
> Thank you for your follow-up and for engaging with our rebuttal. We appreciate you clarifying your remaining concern regarding the ES results in Table 4.
>
> We agree that the lack of significant improvement in the ES for `LR` in Table 4 warrants a detailed explanation. Our argument, supported by both the literature and the new experiment below, is that this observation stems not from a shortcoming of our method, but from a well-documented limitation of the ES itself. As we noted in our initial rebuttal, the ES is known to be relatively insensitive to changes in variance and correlation structure, which are precisely the aspects of the distribution that our post-hoc LR method is designed to correct.
>
> To provide a clear, empirical illustration of this point, we designed a controlled synthetic experiment based on the dataset in Figure 1. The goal here is not to replace the real-world results of Table 4, but to isolate this specific property of the scoring rules in a setting free from the confounding variables of complex, real-world data.
>
> We use an oracle predictor that knows the true data-generating distribution from Figure 1 for everything except the spread around the arc, which is controlled by a standard deviation parameter $\sigma$. We then evaluate the predictor's NLL and ES as we vary its estimate of $\sigma$. The true value is $\sigma=0.05$.
>
> |   $\sigma$ | NLL              | ES     |
> |--------:|:-----------------|:-----------------|
> |    0.01 | 12.01 ± 0.195    | 0.8733 ± 0.00298 |
> |    0.03 | 1.274 ± 0.0222   | 0.8724 ± 0.00267 |
> |    0.04 | 0.9232 ± 0.0129  | **0.8723 ± 0.00295** |
> |    0.05 (True) | **0.8557 ± 0.00870** | 0.8740 ± 0.00171 |
> |    0.06 | 0.8781 ± 0.00629 | 0.8741 ± 0.00182 |
> |    0.07 | 0.9365 ± 0.00483 | 0.8753 ± 0.00214 |
> |    0.10  | 1.161 ± 0.00272  | 0.8741 ± 0.00163 |
> |    0.20  | 1.774 ± 0.00106  | 0.8782 ± 0.00176 |
> *(Metrics averaged over 10 runs, with standard error).*
>
> This experiment strikingly illustrates the issue:
> 1. The NLL shows a sharp, clear minimum at the true value of $\sigma=0.05$, correctly identifying the best model.
> 2. The ES remains almost completely flat for a wide range of $\sigma$ values (from 0.01 to 0.10). It fails to reliably distinguish a model with the correct variance from one that is substantially over- or under-confident.
>
> This insensitivity is so profound that detecting a statistically significant signal with the ES requires an impractically large number of samples. The table below shows that only with **5000 runs** does the ES minimum align with the true $\sigma$, and even then the differences are minuscule:
>
> |   $\sigma$ | NLL              | ES     |
> |--------:|:-----------------|:-----------------|
> |    0.01 | 11.91 ± 0.00773   | 0.8754 ± 0.000139 |
> |    0.03 | 1.263 ± 0.000868  | 0.8751 ± 0.000139 |
> |    0.04 | 0.9177 ± 0.000495 | 0.8751 ± 0.000139 |
> |    0.05 (True) | **0.8487 ± 0.000323** | **0.8750 ± 0.000139** |
> |    0.06 | 0.8732 ± 0.000231 | 0.8751 ± 0.000138 |
> |    0.07 | 0.9329 ± 0.000176 | 0.8751 ± 0.000138 |
> |    0.1  | 1.159 ± 0.000104  | 0.8756 ± 0.000137 |
> |    0.2  | 1.774 ± 6.91e-05  | 0.8797 ± 0.000133 |
> *(Metrics averaged over 5000 runs, with standard error).*
>
> This controlled experiment, therefore, explains the results in Table 4: `LR` is improving the shape and spread of the distribution, but the ES is not the right tool to measure this specific kind of improvement, highlighting the importance of using multiple, complementary evaluation metrics like NLL.
>
> We will add this analysis to the appendix to provide context for our results and guide future work. We hope this addresses your concern and clarifies the interpretation of our findings. We are happy to elaborate further if anything remains unclear.

---

> > ### Comment · Reviewer_qdRU · 2025-08-08
> >
> > Thanks a lot for the detailed explanation, which is supported by further analysis and results.
> >
> > I agree that the results can be used to conjecture the potential reasons for the results in table 4. However, I do believe that you need to justify/validate the proposed methods both on synthetic and real-world datasets. The former of course can give you more insights on why it works or does not work in certain cases; the latter, however, indicates better whether it is gonna be useful in practice (hence the significance of the work).

---

### Official Review · Reviewer_4tSF · 2025-07-03

**Clarity:** 2
**Significance:** 2
**Originality:** 2
**Rating:** 4
**Confidence:** 3

**Summary:**

This paper shows a simple way to make multivariate normalizing flows give better uncertainty by fixing their hidden codes. Instead of tweaking outputs directly, the authors look at the length of each hidden code and remap it to match a known length distribution. They call this latent recalibration (LR). After a model is trained, LR rescales the codes using a quantile map so their spread is correct. The new modle still keeps a clear density formula, runs fast, and has math bounds for coverage. Tests on table data sets and one image set show that LR cuts down calibration error and improves the fit of predicitons. The method also links to other conformal tools and matches standard recalibration in the one-dimensional case.

**Questions:**

Please also see questions in above section.

- for the improved NLL performance, does it mainly come from the base or the jacobian term?

**Ethical Concerns:**

["NO or VERY MINOR ethics concerns only"]

**Final Justification:**

I've raised my score to a weak accept as most of my concerns are well addressed by the rebuttal.

**Limitations:**

yes

**Quality:**

2

**Strengths And Weaknesses:**

Strengths

  - the paper provides definitions (latent calibration, L-ECE) and proves that the recalibrated model has finite-sample coverage guarantees, and unlike sampling-based methods, LR yields a closed-form PDF, so downstream tasks requiring likelihoods can still be done efficiently.

  - this work also addresses a real gap, that existing recalibration methods either only cover univariate outputs or lack an explicit density. LR fills this need for multivariate flows. Also the method can be applied post-hoc to any normalizing flow model with a simple radial transform, without retraining the base model. Assessing calibration in the latent space via norms is a fresh perspective, mapping latent norms by quantiles seems to be effective, and ties neatly to conformal frameworks.

Weaknesses

  - the proposed LR method seems to be inherently tied to normalizing flows, assuming models with an invertible transformation to a known latent distribution. While NFs are a powerful model class, is it possible to extend these ideas to other popular generative models like VAEs or diffusion models, which are now dominant in many domains. The paper would be stronger if it discussed potential pathways or challenges for extending these ideas to other model architectures.

  - authors states that producing a full PDF is a major advantage over set-based CP methods. However, the argument could be made more specific. The authors mention that this is "essential for many applications," but they do not provide specific examples of downstream tasks where having the explicit density is a critical, enabling feature. Adding such examples and evaluations would further help.

  - LR only adjusts the length of latent vectors, not their directions. is it possible that if a flow’s miscalibration comes from skewed or rotated latent structure , simply remapping norms may fail to correct important dependencies? Also, by focusing on the norm distribution, LR assumes all misshapes live in magnitude, not orientation. In practice, does this always hold?

---

> ### Author Rebuttal · Authors · 2025-07-31
>
> The core requirements for `LR` are **(1) an invertible transformation** between the response and latent spaces, and **(2) a latent random variable with a known, tractable density**. This makes Normalizing Flows (NFs) a natural and ideal fit for our method.
>
> You are correct that this formulation does not directly apply to models like VAEs or Denoising Diffusion Probabilistic Models (DDPMs), which lack an exact, tractable inverse mapping. We will add a discussion of this limitation to the paper.
>
> While it is not the main focus of our paper, `LR` is directly compatible with the more recent **Conditional Flow Matching (CFM)** methods, which also learn invertible maps and have a known latent distribution. We performed additional experiments that we will include in the appendix.
>
> **Details on Conditional Flow Matching:**
> Given $x \in \mathcal{X}$, we model the conditional PDF $\hat{f}(y|x)$ using a transformation defined by an Ordinary Differential Equation (ODE), $\frac{d\tilde{y}}{dt} = \hat{v}(t, \tilde{y}, x)$, with a neural network-parameterized vector field $\hat{v}$. Trained via the Conditional Flow Matching (CFM) objective, the model learns to map samples from a latent variable $Z \sim \mathcal{N}(0, I)$ to the target conditional distribution. Forward numerical integration of this dynamic generates samples, while reverse integration performs encoding. To compute the likelihood for $y \in \mathcal{Y}$, we first obtain its latent representation $z = \hat{T}^{-1}(y|x)$. The log-likelihood is then given by the continuous change of variables formula, which integrates along the unique path $\tilde{y}(t)$ that solves the ODE with boundary conditions $\tilde{y}(0)=z$ and $\tilde{y}(1)=y$:
> $$ \log \hat{f}(y|x) = \log f_Z(z) - \int_0^1 \text{Tr}\left(\nabla_{\tilde{y}} \hat{v}(t, \tilde{y}(t), x)\right) dt $$
> The computationally intractable Jacobian trace is efficiently approximated using the unbiased Hutchinson's estimator, enabling practical likelihood computation.
>
> **Results:**
> To avoid large tables, we compare the methods on the largest datasets. The CFMs results are aligned with the NFs results, with `LR` standing out particularly on the L-ECE and NLL metrics, with a good performance on HDR-ECE and ES.
>
> | Metric | Method |  AIR |     BI2 |     BI1 |      WAG |      ME3 |     ME1 |     ME2 |      HO1 |      BIO |     BLO |    CAL |     TAX |
> |:----|:--|-------:|--------:|--------:|---------:|---------:|--------:|--------:|---------:|---------:|--------:|-------:|--------:|
> | L-ECE | `BASE` | 0.179  |  0.285  | 0.123  |  0.118  |  0.263  |  0.263  |  0.257  |  0.0538  |  0.145   |  0.133   | 0.013   | 0.108   |
> | L-ECE | `LR` | 0.0146 |  0.0136 | 0.0112 |  0.0139 |  0.0145 |  0.0141 |  0.0119 |  0.00944 |  0.00601 |  0.00721 | 0.00487 | 0.0042  |
> | NLL | `BASE` | 4.23   | -1.39   | 1.94   |  0.0692 | -0.304  | -0.438  | -0.364  | -0.228   | -0.487   | -1.75    | 0.575   | 1.49    |
> | NLL | `LR` | 3.43   | -3.45   | 1.47   | -0.222  | -2.04   | -2.11   | -1.98   |  0.216   | -0.957   | -2.5     | 0.599   | 1.2     |
> | HDR-ECE | `BASE` | 0.198  |  0.258  | 0.129  |  0.0763 |  0.246  |  0.241  |  0.225  |  0.155   |  0.14    |  0.197   | 0.0209  | 0.134   |
> | HDR-ECE | `HDR-R` | 0.0201 |  0.0225 | 0.0131 |  0.0125 |  0.0146 |  0.0141 |  0.0112 |  0.00919 |  0.00748 |  0.0115  | 0.00615 | 0.00881 |
> | HDR-ECE | `LR`   | 0.0579 |  0.0214 | 0.0198 |  0.0549 |  0.0587 |  0.0514 |  0.036  |  0.112   |  0.0083  |  0.09    | 0.0204  | 0.0407  |
> | ES | `BASE` | 1.52   |  1.1    | 0.807  |  0.898  |  0.701  |  0.7    |  0.703  |  0.835   |  0.744   |  0.75    | 0.881   | 0.827   |
> | ES | `HDR-R`  | 1.53   |  1.07   | 0.798  |  0.888  |  0.671  |  0.675  |  0.685  |  0.796   |  0.733   |  0.742   | 0.882   | 0.811   |
> | ES | `LR`   | 1.51   |  1.03   | 0.795  |  0.89   |  0.644  |  0.643  |  0.649  |  0.833   |  0.725   |  0.735   | 0.881   | 0.818   |
>
> **Practical advantages of producing an explicit PDF**
>
> We designed a new experiment to show a practical scenario where having a calibrated, explicit PDF is beneficial. To avoid redundancy, you can find the details of this experiment in the answer to Reviewer qdRU.
>
> **LR only adjusts the length of latent vectors, not their directions**
>
> You are right, and it is a core design choice of our method.
>
> `LR` intentionally adjusts only the magnitude of latent vectors, not their direction. We acknowledge that this means `LR` cannot fix miscalibration arising from errors in the *orientation* of the learned latent manifold.
>
> However, this focused approach is a principled trade-off that yields significant benefits:
>
> - **Tractability and guarantees:** It simplifies the difficult multivariate calibration problem into a tractable univariate one (calibrating norms). This is precisely what enables us to provide finite-sample guarantees and establish a clean connection to conformal prediction and existing Quantile Recalibration.
> - **Empirical performance:** Our experiments demonstrate that this radial-only adjustment is remarkably effective, suggesting that correcting the norm distribution addresses a major source of miscalibration in many conditional normalizing flows.
>
> We will add a discussion of this deliberate design choice and its implications to the limitations section of the paper.
>
> **Does the NLL performance gain come from the base or the jacobian term?**
>
> To investigate the source of the NLL improvement, we consider the decomposition of the recalibrated NLL:
> $$-\log \hat{f}'(y \mid x) = -\log f_{Z}\left( z \right) - \log \left| \det\left( \nabla_z R(z) \right) \right|^{-1} -\log \left| \det\left( \nabla_y \hat{T}^{-1}(y; x) \right) \right|$$
>  with $z' = \hat{T}^{-1}(y; x)$ and $z = R^{-1}(z')$. The third term is identical for both `BASE` and `LR`.
> We analyze the contributions of the first two terms across the largest tabular datasets to avoid the table being too large.
> All reported terms are averaged over the test set and over 10 runs.
>
> | Model | Expression | SC1 |   AIR |     BI2 |    BI1 |     WAG |    ME3 |    ME1 |    ME2 |    HO1 |     BIO |     BLO |    CAL |    TAX |
> |:-|:--------------------------------------|------:|------:|------:|-------:|-------:|-------:|-------:|-------:|-------:|----------:|-------:|--------:|-------:|
> | `BASE` | $-\log f_{Z}(z)$ | 25.1  | 9.14  |   7.68 |  3.19  |  3.55  |  3.28  |  3.29  |  3.14  |  3.11     |  2.92  |  3.08   | 2.84   | 2.87   |
> | `LR` | $-\log f_{Z}(z')$ | 22.7  | 8.47  |   5.68 |  2.82  |  2.8   |  2.82  |  2.81  |  2.81  |  2.82     |  2.82  |  2.83   | 2.82   | 2.83   |
> | `LR` | $-\log \left\| \det\left( \nabla_z R(z) \right) \right\|^{-1}$ |  1.58 | 0.474 |   0.14 |  0.27  |  0.376 |  0.244 |  0.137 |  0.149 |  0.217    |  0.108 |  0.0512 | 0.0247 | 0.0427 |
> | `BASE` | $-\log \hat{f}(y \mid x)$ | -1.91 | 2.85  | -10.2  | -0.879 | -1.25  | -1.57  | -2.01  | -1.88  |  0.000887 | -1.1   | -3.01   | 0.577  | 1.54   |
> | `LR` | $-\log \hat{f}'(y \mid x)$ | -2.73 | 2.65  | -12.1  | -0.98  | -1.62  | -1.79  | -2.35  | -2.05  | -0.0742   | -1.09  | -3.22   | 0.586  | 1.55   |
>
> The table reveals a clear pattern. The NLL improvement from `LR` is primarily driven by the first term. By radially transforming the latent codes $z'$ to new points $z$ that are more consistent with the base density $f_Z$, the latent density term $-\log f_Z(z)$ is significantly reduced. The recalibration Jacobian (second term) typically adds a small penalty (increases NLL), but this is almost always outweighed by the large gains from the first term. This confirms that `LR` works by finding more "plausible" latent codes for the observed data under the base latent distribution.

---

> > ### Comment · Reviewer_4tSF · 2025-08-06
> >
> > Thank you for the detailed rebuttal. I appreciate the effort you have put into addressing the concerns raised, including running new experiments. The new results and clarifications have largely resolved my concerns and will be reflected in my updated evaluations.

---

### Note · Authors · 2025-08-13

Dear Reviewers and Area Chair,

We sincerely thank you for the insightful and constructive review process. We were especially pleased to hear that our paper brings a "fresh perspective" that "ties neatly to conformal frameworks" (Reviewer 4tSF), is "well-written and very easy to follow" (Reviewer qdRU) and "the empirical evaluations are extensive" (Reviewer 54XG). We believe that our approach provides a significant contribution to the calibration literature, with applications to expressive multivariate probabilistic models.

The review process was particularly valuable in challenging us to demonstrate the practical implications of our work and led to several key additions:
1.  We addressed concerns regarding the applicability of LR to other generative models by showing that LR is directly compatible with **Conditional Flow Matching** models, achieving significant improvements.
2.  We conducted a new **decision-making experiment** to provide a concrete example where having a full, calibrated PDF leads to demonstrably better outcomes than sampling-based approaches.
3.  We have now included experimental results benchmarking the **computational time** of LR against competing methods, confirming its efficiency, especially on large datasets.
4.  We provided a detailed explanation and a new controlled synthetic experiment to clarify why the **Energy Score (ES)** showed little improvement, demonstrating its insensitivity to the specific calibration improvements our method provides, in contrast to the NLL. Following the suggestions of Reviewer qdRU, we provide an additional experiment on real-world datasets. We transform the magnitude of latent vectors using $r'(l) = wr(l) + (1 - w)l$, which reduces to `BASE` when $w = 0$ and `LR` when $w = 1$.
| $w$| 0.0 | 0.25 | 0.5 | 0.75 | 1.0 |
|:-|:-|:-|:-|:-|:-|
| NLL| -0.7299 ± 0.69  | -1.002 ± 0.54   | -1.206 ± 0.43   | -1.338 ± 0.38   | -1.386 ± 0.35   |
| ES | 0.6978 ± 0.0065 | 0.6957 ± 0.0067 | 0.6938 ± 0.0069 | 0.6924 ± 0.0071 | 0.6925 ± 0.0085 |

The results, displayed for the dataset wage and consistent across datasets, show that the NLL shows high sensitivity while the ES is almost completely insensitive.

Once again, we thank all reviewers for their invaluable suggestions and insights, which have significantly contributed to the improvement of our paper.

---

### Decision · Program_Chairs · 2025-09-17

**Decision:**

Accept (poster)

**Comment:**

### Summary of scientific claims and findings

The paper introduces a framework for multivariate calibration and a post-hoc method for conditional normalizing flows. They rescale latent vector norms to match a known distribution, producing a calibrated model with an explicit PDF. The method is theoretically grounded with finite-sample coverage guarantees and links to conformal prediction. Empirical results show improvements in calibration, likelihood, and efficiency.

### Strengths

* Clear and well-written presentation.
* Conceptually simple and novel approach.
* Theoretical guarantees.
* Extensive empirical evaluation.
* Additional experiments during rebuttal addressed concerns about runtime, decision-making utility, and metric sensitivity.

### Weaknesses and missing elements

* Applicability limited to invertible models.
* Improvements on real-world datasets less pronounced compared to synthetic cases.
* Method adjusts latent magnitudes only, leaving potential miscalibrations in orientations unaddressed.
* Some evaluation metrics show limited sensitivity, raising interpretability challenges.

### Key reasons for acceptance

1. The method is simple, yet principled.
2. Produces explicit PDFs.
3. Empirical evaluation is extensive.
4. Limitations are clearly acknowledged and well-documented.

### Summary of discussion and rebuttal

Before rebuttal:

* `4tSF`: Positive on novelty and framing but cautious about applicability beyond normalizing flows. Initially rated borderline, later upgraded to weak accept.
* `54XG`: Found the method clear and evaluations extensive, but questioned efficiency and comparisons; rated borderline accept.
* `qdRU`: Appreciated novelty and explicit PDFs but remained unconvinced by real-world dataset performance; rated borderline accept.

Rebuttal:

* Authors added scaling results, runtime benchmarks, a decision-making experiment, and controlled experiments clarifying Energy Score behavior.

Post-rebuttal:

* `4tSF`: Upgraded to weak accept, concerns largely resolved.
* `54XG`: Maintained borderline accept, noting computational advantages and extensibility.
* `qdRU`: Maintained borderline accept, still cautious about real-world impact but acknowledged novelty and usefulness.

### Weighing these points

All active reviewers leaned toward acceptance after rebuttal. While two remained at borderline, the novelty, theoretical grounding, and practical utility outweigh the limitations.